# A Parkinson's disease CircRNAs Resource reveals a link between circSLC8A1 and oxidative stress

Mor Hanan[1,2], Alon Simchovitz[1,2], Nadav Yayon[1,2], Shani Vaknine[1,2], Roni Cohen-Fultheim[3], Miriam Karmon[3], Nimrod Madrer[1,2], Talia Miriam Rohrlich[2,4], Moria Maman[2,4], Estelle R Bennett[1,2], David S Greenberg[1,2], Eran Meshorer[2,4], Erez Y Levanon[3] iD, Hermona Soreq[1,2,*] iD & Sebastian Kadener[1,5,**] iD

## Abstract

Circular RNAs (circRNAs) are brain-abundant RNAs of mostly unknown functions. To seek their roles in Parkinson's disease (PD), we generated an RNA sequencing resource of several brain region tissues from dozens of PD and control donors. In the healthy substantia nigra (SN), circRNAs accumulate in an age-dependent manner, but in the PD SN this correlation is lost and the total number of circRNAs reduced. In contrast, the levels of circRNAs are increased in the other studied brain regions of PD patients. We also found circSLC8A1 to increase in the SN of PD individuals. CircSLC8A1 carries 7 binding sites for miR-128 and is strongly bound to the microRNA effector protein Ago2. Indeed, RNA targets of miR-128 are also increased in PD individuals, suggesting that circSLC8A1 regulates miR-128 function and/or activity. CircSLC8A1 levels also increased in cultured cells exposed to the oxidative stress-inducing agent paraquat but were decreased in cells treated with the neuroprotective antioxidant regulator drug Simvastatin. Together, our work links circSLC8A1 to oxidative stress-related Parkinsonism and suggests further exploration of its molecular function in PD.

**Keywords** AGO2; CircRNA; CircSLC8A1; Parkinson's disease; RNA sequencing
**Subject Categories** Biomarkers; Neuroscience

## Introduction

Parkinson's disease (PD) is the second most common neurodegenerative disease and the most common movement disorder, affecting 1–2% of the population over 65 (Farrer, 2006; Farrer *et al*, 2006).

Parkinson's disease is characterized by a progressive loss of dopaminergic neurons in the SN, which is attributed to multi-leveled and elusive complex interactions between genetic susceptibilities, male sex, environmental toxins, mitochondrial dysfunction, and imbalanced signaling processes (Farrer, 2006). Consequent depletion of the nigrostriatal pathway and of striatal dopamine then lead to the most prominent motor symptoms of the patients, including bradykinesia, hypokinesia, rigidity, resting tremor, and postural stability. The leading risk factors for PD include age and environmental exposure to herbicides, pesticides, and metal-derived substances (Collier *et al*, 2011). The major molecular hallmark of PD is intracellular accumulation of protein deposits (named Lewy bodies) which are primarily composed of precipitates of the alpha-synuclein (SNCA) protein. Much of the knowledge of genes and pathways involved in PD derives from studies of inherited forms of the disease. Those have been associated with mutations in alpha-synuclein (SNCA), a regulator of synaptic function, as well as Parkin and Pink1, which are involved in mitochondria quality control, and DJ-1, associated with oxidative stress response and the mitochondrial protein kinase LRRK2 [4]. Disease age of onset in families with mutations in these genes is between 20 and 50, considerably earlier than 65 years of age, the typical onset age for the sporadic forms of the disease. Additionally, PD involves massive changes in RNA metabolism (La Cognata *et al*, 2015), both in brain neurons (Liu *et al*, 2017) and in patients' leukocytes (Soreq *et al*, 2008, 2012, 2014; Simchovitz *et al*, 2020), but contributions of circular RNAs (circRNAs) to PD risks have so far remained unexplored.

CircRNAs are a recently rediscovered type of RNA generated by co-transcriptional circularization of specific exons by the spliceosome in a process named back-splicing (Memczak *et al*, 2013; Ashwal-Fluss *et al*, 2014; Jeck & Sharpless, 2014; Li *et al*, 2018; Kristensen *et al*, 2019). They accumulate with age in the brains of flies, nematodes, and mice (Westholm *et al*, 2014; Gruner *et al*, 2016; Cortes-Lopez *et al*, 2018), and some of them are enriched in

1 Department of Biological Chemistry, The Institute of Life Sciences, The Hebrew University of Jerusalem, Jerusalem, Israel
2 The Edmond and Lily Safra Center for Brain Sciences, The Hebrew University of Jerusalem, Jerusalem, Israel
3 Mina and Everard Goodman Faculty of Life Sciences, Bar-Ilan University, Ramat Gan, Israel
4 Department of Genetics, The Institute of Life Sciences, The Hebrew University of Jerusalem, Jerusalem, Israel
5 Biology Department, Brandeis University, Waltham, MA, USA
*Corresponding author. Tel: +972 548820629; E-mail: hermona.soreq@mail.huji.ac.il
**Corresponding author. Tel: +972 548820629; E-mail: skadener@brandeis.edu

synapses and dendrites (Rybak-Wolf et al, 2015; Veno et al, 2015; You et al, 2015). Most of the known circRNAs are derived from RNA-polymerase II transcription of protein coding genes, and their circular structure makes them resistant to cellular exonucleases and hence exceptionally stable RNA molecules compared to canonical mRNA (Jeck et al, 2013). Hence, circRNAs can potentially become biomarkers for disease and/or the follow-up of treatment efficacy or targets for new therapeutics (Zhang et al, 2018). While thousands of circRNAs have been described, the function of only very few has been elucidated. These include circCDR1as, the most abundant mammalian circRNA. CircCDR1as originates from the antisense strand of the cerebellar degeneration-related protein 1 (CDR1) gene which is highly specific to neurons (Hansen et al, 2013). It binds the miR effector protein Argonaute2 (Ago2) and harbors 73 binding sites for miR-7 (Hansen et al, 2013; Memczak et al, 2013). Initially, it was though that CDR1as sponges and degrades miR-7. However, knockout of CDR1as results in lower, not higher levels of this miRNA, suggesting that CDR1as regulates the function and/or stability of miR-7 in a more complex way (Piwecka et al, 2017). This process likely involves the non-coding RNA Cyrano that regulates miR-7 levels and activity (Guo et al, 2014; Kleaveland et al, 2018).

Notably, several circRNAs have been shown to be functional in vivo in mice and flies (Holdt et al, 2016; Chen et al, 2017; Li et al, 2017a,b; Pamudurti et al, 2017). Those circRNAs have been proposed to have a variety of molecular functions including RBP binding and transport (Ashwal-Fluss et al, 2014), rRNA maturation, and miRNA stabilization (Holdt et al, 2016). Moreover, a subset of circRNAs is translated but no function has been found for their protein products (Legnini et al, 2017; Pamudurti et al, 2017; Yang et al, 2017; Liang et al, 2019). Interestingly, several reports have suggested a link between circRNAs and neurodegenerative disorders including Alzheimer's disease and amyotrophic lateral sclerosis (Lukiw, 2013; Errichelli et al, 2017; Shi et al, 2017; Dube et al, 2019). Supporting this notion, many circRNA features, including their age-dependent accumulation in the brain, suggest potential roles of these RNAs in neuronal demise. Together, this called for establishing a resource of circRNAs in the healthy and PD brain regions where the potential role of these intriguing transcripts in disease could be addressed.

CircRNAs production has been shown to be regulated by RNA editing or hyper-editing events—higher levels of editing beyond transcriptome alignment (Porath et al, 2014; Ivanov et al, 2015; Rybak-Wolf et al, 2015). As 99% of the A-to-I RNA editing in humans occurs in Alu elements, those elements may serve as a measure of global editing levels (Levanon et al, 2004). Alu elements are the most abundant transposable elements in the human genome (over one million copies), and RNA originated from Alu elements inserted in opposite orientations can form dsRNA structures, providing a substrate for RNA editing by adenosine deaminases (ADAR proteins). Interestingly, many circRNAs are flanked by introns that contain Alu or other repeats which have been postulated to mediate the generation of some circRNAs (Ivanov et al, 2015). Multiple Alu pairing events in introns suggest a role of competition between Alu pairing and the formation of alternative circRNAs from the same locus (Zhang et al, 2014a,b). Supporting this notion, ADAR expression negatively correlates with circRNA levels in neural tissue in flies as well as in mammalian cell lines (Ivanov

et al, 2015; Rybak-Wolf et al, 2015). However, the regulation is more complex. On one hand, the relative location of the Alu elements (or other repeats) is key for determining how ADAR modulates circRNA biogenesis (Ivanov et al, 2015). On the other hand, this regulation also involves DHX9, an RNA helicase which binds inverted-repeat Alu elements, unwinds their secondary structure and represses circRNA abundance (Aktas et al, 2017). However, whether such editing events relate to circRNA abundance in human healthy or diseased brain tissues has not yet been investigated.

To address potential roles of circRNAs in PD and determine whether these molecules could biomark this disease, we comprehensively profiled the circRNAs in three brain regions from dozens of PD and healthy individuals: The substantia nigra (SN), where much of the PD-related demise of dopaminergic neurons takes place, the medial temporalis gyrus (MTG), and the amygdala (AMG), which controls acute stress responses that are known to accompany this disease. Comparing brain tissues from human PD patients to matched apparently healthy aged individuals identified a PD-related loss of the age-dependent increase of SN circRNAs and specific changes in RNA editing of Alu elements. Having found that circSLC8A1 is significantly upregulated in the SN of PD individuals, we selected this circRNA as an in-depth example for addressing the role of circRNAs in the PD brain as it relates to the main PD-inducing risks such as oxidative stress. In sum, our work provides an important resource for studying circRNAs in PD and demonstrates that these RNAs are miss-regulated in the brain of PD individuals.

## Results

### A genomic resource for studying circular RNAs in PD brains

To determine how gene expression at large and circRNA profiles in particular are changed in specific brain regions from individuals with PD, we generated and sequenced dozens of rRNA-depleted RNA-seq libraries from the AMG, SN, and MTG of PD or healthy individuals (Fig EV1A–D; see Dataset EV1 for clinical data for the donors). The sequenced and analyzed tissues included 27 control and 42 PD samples (libraries were only prepared from samples with RIN > 6.5, and we excluded samples with low sequencing number of circRNAs; Appendix Fig S1E–H). This relatively large set assisted in dealing with the individual heterogeneity characteristic of human brain tissues (Barbash et al, 2017). We sequenced these libraries at a deep level (50 M reads per sample on average; Fig EV1I, Dataset EV2), allowing reliable and simultaneous detection and quantification of mRNAs, long non-coding RNAs (lncRNAs) and circRNAs. We then used a bioinformatics pipeline to identify and annotate circRNAs and mRNAs (Memczak et al, 2013).

An initial analysis of the heterogeneity of the samples by non-supervised hierarchical clustering indicated substantial clustering of the MTG or AMG samples by brain region but failed to distinguish the PD from the control samples (Figs 1A and B, and EV2A). This likely reflected heterogeneity between individuals; the pronounced brain region specificity of gene expression patterns that dominate this analysis over other factors such as gender, age, Braak stage differences, disease symptoms severity, and possibly the impact of treatment by diverse medications may also be relevant (Braak et al, 2003). Unlike the MTG or AMG samples, non-supervised clustering

analysis of the SN enabled better separation between the gene expression patterns of PD and control individuals (Figs 1C and EV2B–D). This might reflect the bigger differences in gene expression due to the loss of dopaminergic neurons in the SN. Correspondingly, tyrosine hydroxylase (TH) levels were significantly decreased in the PD SN samples (t-test $P = 0.025$; Fig 1D), compatible with the major decrease in dopaminergic metabolism in this brain region upon PD.

Global gene expression following correction for cell composition (for microglia, astrocytes, and neurons, see Materials and Methods) indicated that the differences in the SN of PD individuals included genes associated with PD (e.g., SNCA, TH, DNAJC6, SYNJ1, GBA, DNAJC6, and SLC6A3) (corrected $P = 0.00048$), synapse (corrected $P = 2.06E-05$) and neurodegeneration (corrected $P = 0.0017$; Miller *et al*, 2004; Zhang *et al*, 2005; Moran *et al*, 2006; Simunovic *et al*, 2009; Lewis & Cookson, 2012). Gene Ontology (GO) enrichment analysis of the mRNAs differentially expressed (DE) between PD and control individuals indicated that the DE mRNAs in the SN of PD patients are enriched for genes involved in cell junctions, synaptic vesicle (corrected $P = 8.52E-04$), ion transport (corrected $P = 1.37E-04$), and dopamine biosynthetic processes (corrected $P = 0.0198$), (Fig 1E, Dataset EV3 for all DE genes, FDR < 0.01 that were used for the GO term analysis, Dataset EV4 shows detailed the identified GO terms). We next utilized single-cell RNA sequencing data from human brains to determine the cell types expressing these DE genes (McKenzie *et al*, 2018). We found most of the DE transcripts (Fig 1F and G) to be expressed in several cell types, with endothelial cells, oligodendrocytes, and astrocytes expressing a higher proportion of them than neurons and microglia. Also, the SN neuron-specific DE genes predictably showed pronounced decreases in PD (63 vs. 14 genes, chi-square test, $P = 3.5E-12$), whereas oligodendrocytes and microglia-expressed genes showed a tendency for increases (24 and 12 vs. 2 and no genes in healthy brains), possibly reflecting their increased fractions and/or elevated inflammation.

To seek putative co-regulation of specific DE mRNAs, we performed weighted gene correlation network analysis (WGCNA; Langfelder & Horvath, 2008). This analysis identified transcripts with closely altered expression patterns, creating gene modules with intra-related variability scores (Fig EV2E and F and Dataset EV5). Genes were clustered into 15 modules indicated by different colors (Fig 1H, left color scale), and we assessed their relationships to sample traits such as sex, age, brain region, and condition by calculating the correlation and $P$ value for each module and trait (Fig 1H). We also measured correlations between module memberships and the closeness of each gene's significance in particular modules. Significant $P$ values emerged in certain module–trait relationships, including the brown and red modules with the sample condition (PD or control), the blue module with tissue classification, and the cyan module with the sex of the individual (correlation = 0.94 $P < 1e-200$, Fig 1I). The latter module included many sex-related genes like DDX3Y (DEAD-box helicase 3, Y-linked), XIST (X inactive-specific transcript), NLGN4Y (neuroligin 4, Y-linked), and others (correlation = 0.99 $P = 1.7e-30$, Fig 1I). Pathway analysis and a search for enriched GO terms for tissue-related modules further found various neuronal and synaptic function terms (Fig 1J, blue) and linked disease-associated modules with transcriptional regulation, RNA processing, and mitochondria (cor = 0.56 $P = 3.7e-51$, cor = 0.56 $P = 2.4e-70$, Fig 1I and J, red and brown,

respectively, detailed GO terms in Appendix document). Compatible with this outcome, we found several DE splicing factors in PD tissues (Fig EV2G and H), supporting reports of aberrant and alternative splicing in several brain diseases, including PD (Soreq *et al*, 2012, 2014; Fu *et al*, 2013; La Cognata *et al*, 2015). As several splicing factors can regulate circRNA production (Ashwal-Fluss *et al*, 2014; Conn *et al*, 2015; Errichelli *et al*, 2017; Yu *et al*, 2017), these results motivated us to determine potential changes in circRNA levels in the brain of PD individuals.

## The SN presents a unique PD-influenced circRNA profile

Exon circularization reflects a choice between linear and back-splicing processes (Fig 2A). To identify circRNAs which are altered in PD, we utilized two different circRNA pipelines (see Materials and Methods; Li *et al*, 2017a,b; Pamudurti *et al*, 2017; preprint: Rabin *et al*, 2019) to identify and quantify circRNAs in all of the sequenced samples. Interestingly, we found 3,407 different circRNAs that are only expressed in PD brains, whereas 1,028 emerged as unique to healthy controls (Fig 2B), suggesting disease-related changes in the back-splicing process. Notably, 2,755 of all circRNAs emerged in all studied brain regions (Fig 2C), but 2,914, 3,918, and 2,483 circRNAs were uniquely expressed in the AMG, SN, and MTG, respectively (Fig 2C). This indicated distinct regulation over circularization of specific transcripts in healthy and PD brain tissues. Supporting this notion, non-supervised hierarchical PCA clustering segregated the SN circRNAs from those of the AMG and MTG, which were merged together (similar to what we observed with mRNAs; Fig 2D, and Appendix Fig S3A). Also, as much as 26% of the detected circRNAs but only 2% of the mRNAs were unique to the SN (Fig 2E and F, gray), whereas 19% of the identified circRNAs but 82% of the mRNAs were shared between all tissues (Fig 2E and F, green). To rule out that this reflects the presence of many circRNAs expressed at low levels (which might be regarded as noise), we eliminated from the analysis those circRNAs expressed at very low levels (< 10 reads). This correction indeed separated the SN even further from the other brain regions (raising the fraction of circRNAs which are unique to the SN from 26% to 39%, see Materials and Methods for details). Nevertheless, we found no clustering by age (Fig EV3B), disease condition (Fig EV3C), or sex (Fig EV3D), and no significant differences between females and males in total circRNA abundance (Fig EV3E). Together, these findings indicate that circRNAs are brain region-specific and potentially attribute their unique patterns in the SN to the loss of dopaminergic neurons in this disease-related brain region. Importantly, cell-type expression analysis for the host genes of SN circRNAs highlighted genes which are mainly expressed in neurons, oligodendrocytes, and astrocytes (Fig 2G), suggesting that the circRNAs are expressed in several cell types in this tissue.

In healthy donors, the SN expressed higher total numbers of circRNAs compared to the MTG and the AMG (normalized to library total reads, Fig 3A, see Materials and Methods for details). This suggests that cells in the SN possess a particular regulation pattern of circRNA biogenesis. When compared to healthy donors, we have further observed increased numbers of circRNAs in the MTG and AMG of the PD patients (t-test $P = 0.026$ and $P = 0.022$ for MTG and AMG; Fig 3A). However, the SN of PD patients showed lower circRNA numbers than in the SN of healthy controls (t-test $P = 0.02$, Fig 3A), which correlated with and could be secondary to the

neuronal cell loss in the SN of PD patients and/or to altered splicing events in this brain region.

RNA editing has been shown to influence exon circularization in multiple animal models (Ivanov et al, 2015; Rybak-Wolf et al, 2015)

and brain diseases (Gal-Mark et al, 2017; Lorenzini et al, 2018), but it is unknown whether it contributes to circRNAs production in the diseased human brain. To address this issue, we quantified the level of RNA hyper-editing and Alu editing index in the PD brain. Briefly,

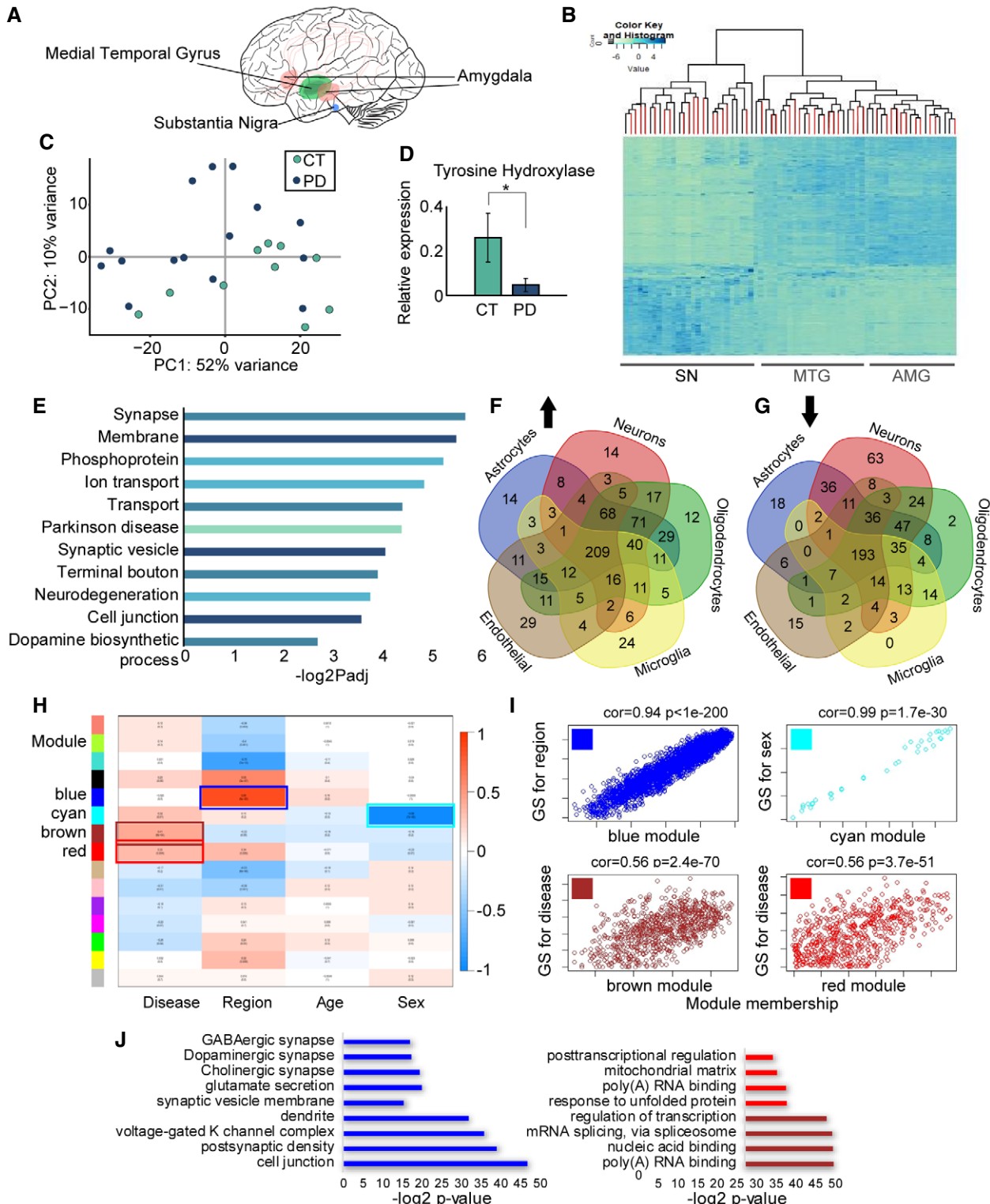

Figure 1.

**Figure 1.  Brain region-specific mRNA profiles indicate multicellular PD-related processes.**

A   Brain region origin of tissues: SN (blue), MTG (green), and AMG (red).
B   Non-supervised clustering heatmap plot indicates substantial clustering of transcripts from each tissue but not of transcripts from patients compared to controls. PD samples indicated in red.
C   PCA clustering of all RNA molecules in the SN control (green) and PD samples (blue).
D   qPCR validation of TH levels in SN samples of PD and controls, normalized to beta-actin mRNA. t-test *P = 0.025. Data presented as mean ± SD. n = 18 for CT and 24 for PD.
E   Top enriched GO pathways of DE genes in the SN of PD vs controls and the corresponding logP values, Wald test (DEseq2 analysis).
F, G  Venn diagrams demonstrating cell type-specific DE genes in the SN of PD vs control tissues, up/down arrows indicate gene groups that were up/down regulated.
H   Module–trait relationship of WGCNA analysis showing correlation and corrected P values (in brackets) for each gene module (indicated by colors) as related to the different external traits selected (disease condition, brain region, age, and sex). Correlation test p value calculated by WGCNA package.
I   Module membership vs. gene significance in selected modules as reflecting external sample traits: disease condition (reflected in the brown and red modules), brain region (the blue module), and sex (the cyan module), regression-based p value calculated by WGCNA analysis. n = 8 for amygdala control, 15 for amygdala PD, 8 for MTG control, and 13 for MTG PD, 10 for SN control and 15 for SN PD.
J   Significant GO terms that emerged as associated with the blue, red, and brown modules and the corresponding log P values, Fisher's exact P value.

we computed an *Alu* editing index, representing the weighted average editing level across all expressed *Alu* sequences in all RNA samples (Roth *et al*, 2019, see Materials and Methods for details). This analysis revealed lower overall editing levels in the SN compared to the other brain regions (Figs 3B and EV4A and B), contrasting the higher expression of circRNAs in this brain region (Fig 3A). These differences are the outcome of RNA editing activity, as the observed mismatches were mainly A-to-G conversions (Fig EV4C). Correspondingly, we found decreased *Alu* editing index in the AMG and MTG tissues from PD patients, but not in the SN tissues compared to controls (Wilcoxon, corrected $P = 0.03$, 0.06 and 0.18 for amygdala, MTG and SN, respectively, Figs 3B and EV4A and B).

The observed decrease in editing in the AMG and MTG tissues of PD patients was anti-correlated with the total increase in circRNAs upon the emergence of PD in these tissues. Moreover, the editing levels showed a similar trend also when assessing them only in circRNAs and in the introns flanking the circularizable exons (for circRNAs corrected $P = 0.025$ and 0.040). However, this was only the case for the AMG and MTG, but not for the SN ($P = 0.3$). For flanking introns, the significance levels were $P = 0.02$ and 0.040 for the AMG and MTG and 0.34 for the SN (Wilcoxon test, Figs 3C and EV4D). Also, circRNA abundance showed a negative correlation with total RNA editing levels in individual samples (correlation = $-0.37$, $P = 0.0015$, Fig 3D). Moreover, we observed differences in RNA editing between the samples originated from healthy and diseased individuals (Fig EV4D). Nevertheless, the *Alu* editing index emerged as non-correlated with age and not affected by sex (Appendix Fig S1A–C). Intriguingly, exploring different tissues separately demonstrated lower levels of ADAR1 in the SN compared to the other tissues (Fig 3E), although ADAR1 levels did not change from control to PD samples (Fig 3E). We conclude that RNA editing and RNA circularization are anti-correlated in several regions of PD and control brains, although we cannot conclude a causal link between these two events.

Previous reports show that circRNAs accumulate in the CNS in an age-dependent manner in flies, worms, and mice (Westholm *et al*, 2014; Gruner *et al*, 2016; Cortes-Lopez *et al*, 2018). To test, for the first time if this is also the case in human brains, we sought trends of age-related accumulation of circRNAs. We found such a trend in the healthy SN (correlation = 0.68, $P = 0.032$) (Fig 3F). Surprisingly, we did not observe positive correlation in any of the other two assayed tissues ($P = 0.81$ and $P = 0.91$ for CT and PD in

AMG and $P = 0.27$ and 0.65 for CT and PD in MTG, Appendix Fig S2A–D), but this could be due to the limited age range of the assayed samples. These differences could be due to differences in overall synthesis or degradation rates of circRNAs in the SN. Further, the observed correlation between age and circRNA levels in the SN could not be detected in the diseased tissues (correlation = $-0.14$ $P = 0.62$, Fig 3G), suggesting that the particular age-associated regulation of circRNAs in the SN reflects a change in the dopaminergic neurons which were largely lost in PD, although secondary effects cannot be ruled out. Together, these results propose that circRNAs are uniquely and strongly regulated in the SN in comparison to the other tested brain regions and suggest potential aging-related roles for circRNAs in the SN which might be impaired upon PD.

Seeking DE circRNAs in all brain tissues from PD and healthy individuals identified 24 DE circRNAs (corrected $P < 0.05$; Fig 3H, Dataset EV6). Those included circSLC8A1 (originated from the $Na^+/Ca^{2+}$ Exchange Protein 1), circNTRK2, and circNTRK3 (hosted by the neurotrophic receptor tyrosine kinase 2 and 3 genes), and circZHX3 (zinc fingers and homeoboxes 3). The significant increase in the PD-modified circSLC8A1 (Fig 3H) raised our interest, since SLC8A1 has recently been identified as a potential factor contributing to neurodegeneration in a mouse model of PD (Sirabella *et al*, 2018).

## CircSLC8A1 is upregulated in PD brains

CircSLC8A1, which is significantly upregulated in the PD SN (Fig 4A and B), originates from the neuronally expressed SLC8A1 gene, which encodes a $Ca^{2+}/Na^+$ transporter that contributes to balancing cytoplasmic $Ca^{2+}$ levels and the surveillance over $Ca^{2+}$-dependent neurotransmission (He *et al*, 2009). Importantly, there are no intronic inverted repeats in the proximity (5 kb upstream and downstream) of the circularizable exon (data not shown). Unlike the increases of circSLC8A1 in the PD brain ($P = 0.025$, Fig 4B), the levels of the linear SLC8A1 mRNA showed no significant change ($P = 0.118$, Fig 4C), as we validated by qRT–PCR from a larger pool of 18 controls and 24 PD SN samples (some of which were not sequenced). qPCR tests further confirmed the increase in circSLC8A1 levels in the PD SN (t-test $P = 0.025$; Fig 4D, in purple) but did not find changes in the host mRNA (Fig 4D, in gray). Further, we did not observe significant differences in the levels of circSLC8A1 between males and females (t-test $P = 0.583$ for control

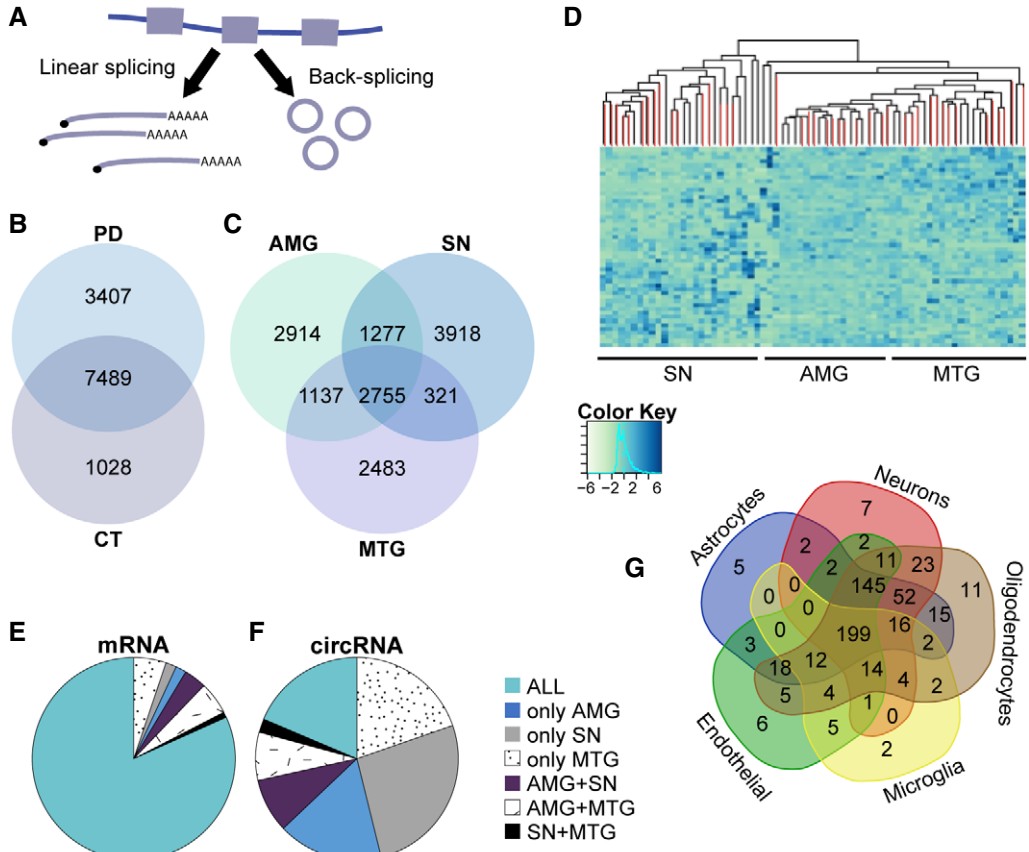

**Figure 2. CircRNA abundance is brain region-dependent and PD-modified.**

A   mRNA processing either involves unidirectional splicing, creating linear mRNA transcripts, or back-splicing generating circRNAs.

B, C   Venn diagram (not to scale) showing brain region- and disease-characteristic circRNA abundance patterns in analyzed brain regions and disease state.

D   Heatmap of circRNA abundance, indicating significant tissue-based SN clustering, PD samples indicated in red.

E, F   Pie chart demonstrating brain-specific expression of mRNAs (E) in comparison to circRNAs (F).

G   Venn diagram demonstrating predicted cell-type characteristics of circRNA host genes in the SN.

and $P = 0.262$ for PD) (Fig 4E). Interestingly, the PD-related increase in this transcript was largely restricted to the SN yet was not due to the higher expression of circSLC8A1 in this tissue, as other brain regions showed yet higher levels of circSLC8A1 (Appendix Fig S3A). Moreover, the levels of both the circular and the linear mRNA SLC8A1 transcripts were strongly correlated in healthy SN samples ($R^2 = 0.47$, $P = 0.0017$). This correlation was completely lost in the SN of PD individuals (Fig 4F and G; $R^2 = 0.016$, $P > 0.05$), possibly indicating disease-induced decline of the regulation over circSLC8A1 production and/or destruction.

**Oxidative stress elevates neuronal circSL8A1 and reduces the SLC8A1 protein**

The host SLC8A1 gene encodes a known sodium/calcium exchanger (Khananshvili, 2013), and thereby contributes to the regulation of cytoplasmic $Ca^{2+}$ levels and $Ca^{2+}$-dependent cellular processes such as neurotransmission (He *et al*, 2009). During neuronal differentiation of embryonic stem cells, we found upregulation of both circSLC8A1 and the linear SLC8A1 mRNA (Appendix Fig S3B). Likewise, published datasets show

upregulated linear SLC8A1 expression in iPSCs differentiation from healthy fibroblasts into neurons. However, the levels of circSLC8A1 remained unchanged in control and PD fibroblast samples (Schulze *et al*, 2018; Appendix Fig S3C). This could indicate that under normal growth conditions, the regulation over circularization and the balance between circSLC8A1 and SLC8A1 mRNA do not change in non-neuronal cells, even in the case of a genetic PD background. Alternatively, we considered that environmental contribution might be more relevant for neuronal differentiation. Specifically, that both circSLC8A1 and the linear SLC8A1 mRNA are upregulated during neuronal differentiation could suggest that changes in circSL8A1 expression in the PD brains might be related to cellular insults or challenges reported to selectively occur in dopaminergic neurons, such as oxidative stress, a known hallmark of PD (Farrer, 2006; Burbulla *et al*, 2017). We therefore tested if the levels of circSL8A1 are directly modulated by oxidative stress. For this purpose, we exposed cultured human SH-SY neuronal cells to increasing concentrations of the oxidative reagent Paraquat (PQ) known to increase the risk of PD (Berry *et al*, 2010). Strikingly, PQ exposure selectively induced dose-dependent increases in circSLC8A levels, but not other circRNAs

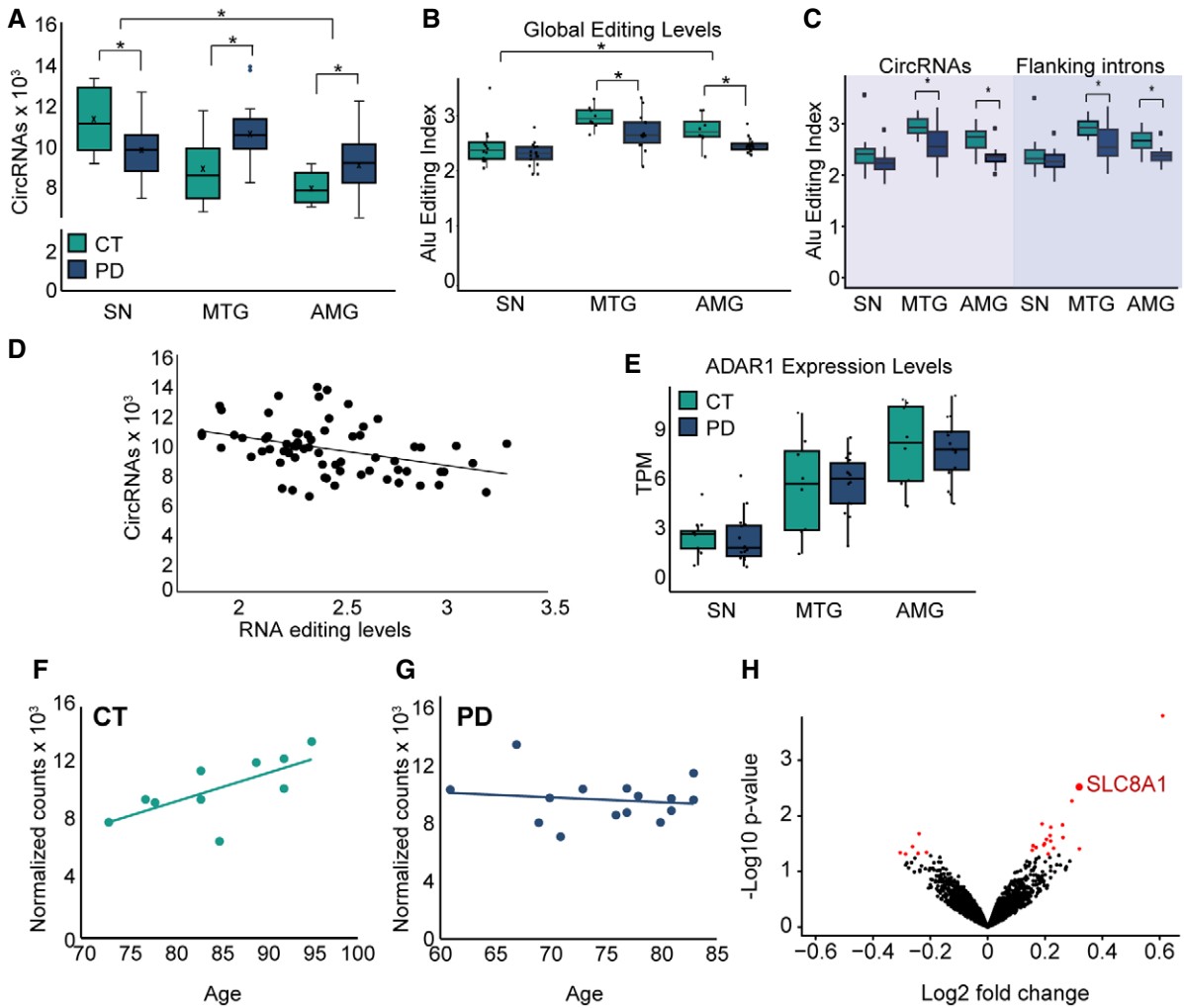

**Figure 3. CircRNA abundance is reduced in the PD SN and negatively correlates to RNA editing levels.**

A   Total circRNAs detected in the SN, MTG, and AMG of both PD and control donors, *P = 0.02 *P = 0.026 and *P = 0.022 for comparison between PD and CT in SN, MTG, and AMG. t-test for total SN vs MTG *P = 2.27E-8 and SN vs AMG *P = 5.86E-10. n = 8 for amygdala control, 15 for amygdala PD, 8 for MTG control and 13 for MTG PD, 10 for SN control and 15 for SN PD. The box is drawn from Q1 to Q3 with a horizontal line drawn in the middle to denote the median and x marks the average. Whiskers mark minimum or maximum values. P value was calculated by t-test.

B   Global editing levels in the 3 brain regions are based on the *Alu* editing index, representing the weighted average editing level across all expressed *Alu* sequences, Wilcoxon, corrected *P = 0.038, 0.055 and 0.42 for amygdala, MTG and SN. Wilcoxon test between the Alu editing index of control and PD samples.

C   Global RNA editing levels based on the *Alu* editing index, studied only in *Alu* elements within circRNA exons and their flanking introns, in the 3 brain regions, for circRNAs corrected *P = 0.025 and 0.040 for the AMG and MTG and P = 0.3 for SN. For flanking introns, *P = 0.02 and 0.040 for AMG and MTG and P = 0.34 for SN, Wilcoxon test. n = 8 for amygdala control, 15 for amygdala PD, 8 for MTG control and 13 for MTG PD, 10 for SN control and 15 for SN PD. The box is drawn from Q1 to Q3 with a horizontal line drawn in the middle to denote the median and x marks the average. Whiskers mark minimum or maximum values.

D   Negative correlation between total RNA editing and circRNA abundance, correlation = −0.37, correlation test p value = 0.0015.

E   Adar1 expression levels in all tissues. The box is drawn from Q1 to Q3 with a horizontal line drawn in the middle to denote the median and x marks the average. Whiskers mark minimum or maximum values.

F, G   Correlations between circRNAs detected in each sample and the age of the control (green, correlation = 0.68, P = 0.032) and PD donors (blue, correlation = −0.14, P = 0.62).

H   Volcano plot of DE circRNAs in PD vs control tissues, red dots indicate statistically significant DE circRNAs according to FDR correction of Wald test (DEseq2 analysis). CircSLC8A1 is marked is red.

(t-test P = 0.028 and P = 0.0215 for 25 μM and 50 μM, respectively, Fig 4H), implying that oxidation *per se* might increase circularization of circSLC8A1 in neurons or decrease circRNA degradation. Notably, we observed a parallel dose-dependent decrease in the levels of the SLC8A1 protein, despite constant

levels of the SLC8A1 mRNA (Fig 4I and J), possibly reflecting faster degradation rate and/or lower translation rate of this protein within the tested 24 h.

Given the strong effect of PQ on circSLC8A1 levels, we wondered whether two known PD protective agents, the cholesterol-reducing

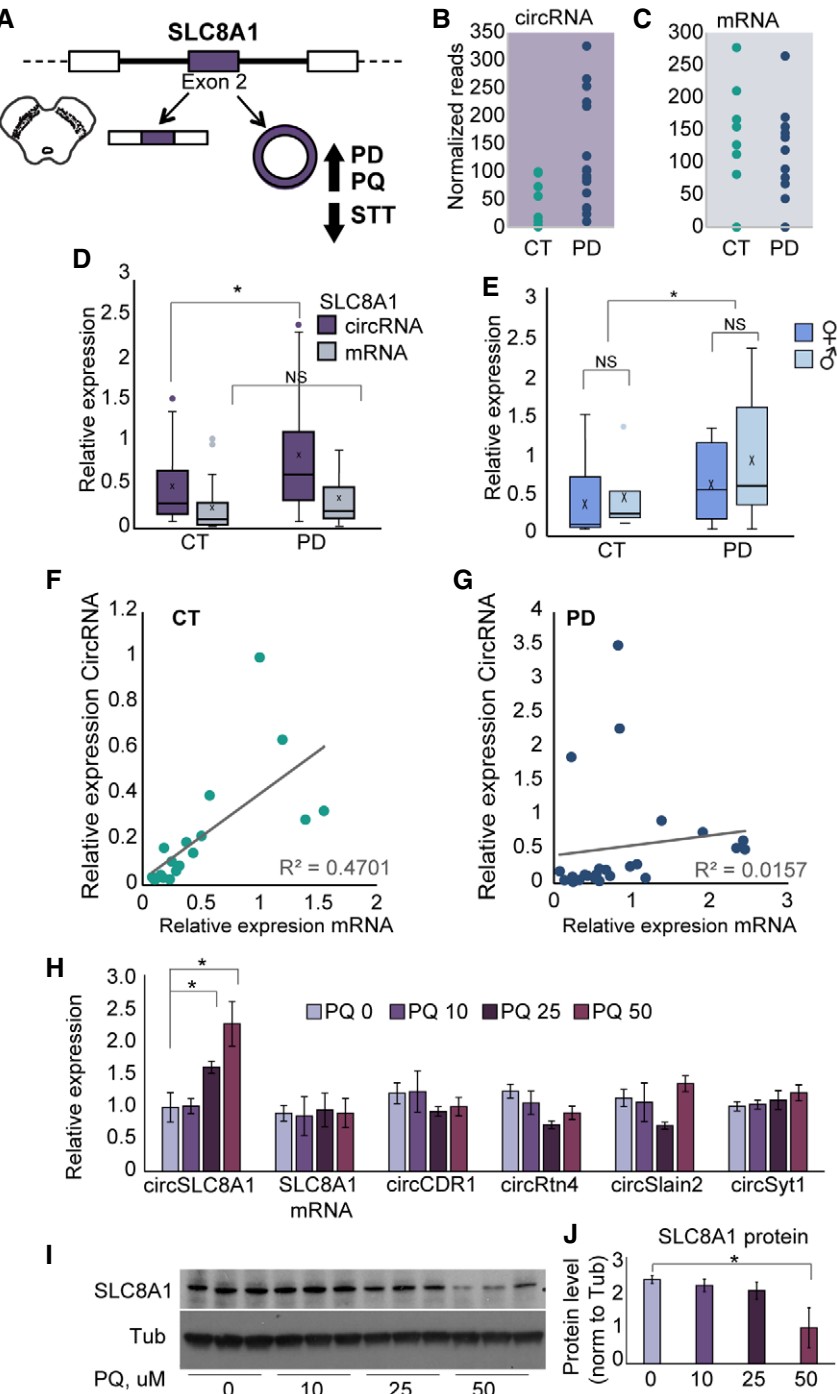

**Figure 4. CircSLC8A1 increases in the PD brain and upon PQ exposure are accompanied by suppression of the SLC8A1 protein.**

A    Graphic representation of circSLC8A1 and its post-transcriptional regulation in PD and under PQ exposure.

B, C   CircSLC8A1 and linear SLC8A1 levels in the SN of control and PD brains, t-test P = 0.025, P = 0.118, respectively.

D    qPCR validations of circSLC8A1 and SLC8A1 mRNA changes in the SN, t-test *P = 0.025. n = 18 for CT and 24 for PD, x defines mean values.

E    No difference in circSLC8A1 levels in females and males, t-test P = 0.583 for control and P = 0.262 for PD. n = 15 females and 16 males.. The box is drawn from Q1 to Q3 with a horizontal line drawn in the middle to denote the median and x marks the average. Whiskers mark minimum or maximum values.

F, G   CircSLC8A1 and linear SLC8A1 mRNA levels are correlated in control, $R^2$ = 0.47 t-test P = 0.0017 but not in PD samples, $R^2$ = 0.016, P > 0.05. n = 18 for CT and 23 for PD. Data presented as mean ± SD. Note that y-axis scales are different to allow better visualization.

H    Increased CircSLC8A1 RNA levels in PQ-exposed SH-SY cells (normalized to Tubb3 and RPL19) while other circRNAs remain unchanged, t-test *P = 0.028 and *P = 0.0215 for 25 μM and 50 μM, respectively. n = 4 biological replicas for each condition. Data presented as mean ± SD.

I, J   Immunoblot decline (Western blot and quantification) of the SLC8A1 protein in PQ-exposed SH-SY cells, with Tubulin as a loading control, t-test *P = 0.015. n = 3 biological replicas for each condition. Data presented as mean ± SD.

    

agent Simvastatin (Yan *et al*, 2018) or the known LRRK2 inhibitor PF-06447475 alter circSLC8A1 in the opposite way. Indeed, these two neuroprotective agents strongly reduced circSLC8A1 (*t*-test $P = 0.007$ for PF-06447475 and $P = 0.047$ for statins, Fig 5A). Additionally, we observed decreased levels of the linear SLC8A1 mRNA (*t*-test $P = 0.01$ for PF-06447475 and 0.0136 for statins; Fig 5A) which resulted in diminished levels of the SLC8A1 protein in SH-SY cells treated with statins (Fig 5B and C). Moreover, immunostaining revealed lower cytoplasmic expression of the SLC8A1 immune-labeled protein in the presence of statins (*t*-test $P = 0.027$ for statins and $P = 0.011$ for PF-06447475, Fig 5D, Appendix Fig S3D). We conclude that both SLC8A1 and circSLC8A1 levels decline in neuro-protected cells.

**CircSLC8A1 binds Ago2 and might regulate miR-128 targets**

The exon which forms the human circSLC8A1 transcript is part of an open reading frame. Hence, it is not surprising that this exon is evolutionarily conserved: We observed 88% and 87% sequence conservation between the human, mouse, and rat genome, respectively (Fig 6A). Cellular sub-fractionations followed by qRT–PCR showed that circSL8A1, like most circRNAs, localizes to the cytoplasm (*t*-test $P = 0.005$ for circSLC8A1, $P = 0.009$ for the nuclear paraspeckles forming lncRNA NEAT1, and $P = 0.009$ for the Rpl19 mRNA, Fig 6B).

A recent report shows that in murine cardiomyocytes, circSLC8A1 binds to the miRNA effector protein Ago2 (Siede *et al*, 2017). To find out if this is also the case in neurons, we tested whether circSLC8A1 is associated with Ago2 by immunoprecipitation followed by qRT–PCR in SH-SY cells (Appendix Fig S3E). Indeed, we found that the human circSLC8A1 interacts with Ago2 also in these cells (Fig 6C). Impressively, the Ago2-immunoprecipitated material reached comparable enrichment of circSL8A1 to that observed for the circCDR1as ($P = 0.029$ for circSLC8A1 and $P = 0.001$ for circCDR1as, Fig 6C, Appendix Fig S3F). Moreover, published Ago2-CLIP data from mouse and human brain tissue (Zhang & Darnell, 2011; Boudreau *et al*, 2014) showed Ago2 binding to several regions of the exon that generates circSLC8A1, with 6 short sequences in this human exon and 5 in the mouse one, 3 of which carry identical sequences (Fig 6A and D—in purple, Dataset EV7). This strikingly conserved binding pattern seems to be restricted to the exon harboring the circRNA, as there is only one additional binding site for Ago2 in the human version of this gene. Interestingly, there are 11 Ago2 binding sites within the SLC8A1 murine 3′ UTR from the analyzed CLIP data, but no sites within the open reading frame of this gene (except for the ones found in the circularizable exon, see above). All of the above strongly suggest that these Ago2-bound regions reflect the binding of the RISC complex to circSLC8A1. Moreover, re-analysis of the Ago2-CLIP reads identified the specific back-splicing junction sequence among those reads, further validating the direct interaction between Ago2 and circSLC8A1 (Dataset EV5). So, while most of the AGO2-CLIP reads cannot distinguish between binding of AGO2 to the linear or circular SLC8A1 RNA isoforms, the circRNA-specific CLIP reads, and the AGO2-IP qPCR unequivocally demonstrate that circSLC8A1 is an Ago2-bound RNA.

It has previously been proposed that circRNAs could operate as miRNA sponges, although there is little evidence that this is the case

for most circRNAs. Given our results showing the interaction of the exon generating circSLC8A1 with Ago-2, we utilized TargetScan (Agarwal *et al*, 2015) to identify potential miRNA binding sites within circSLC8A1 (Fig 6E). We found 7 potential target sites for miR-128. Three of these sites have been identified as Ago2-bound in available human CLIP experiments (Fig 6D, in green), (Boudreau *et al*, 2014). Further, we wished to test if circSLC8A1 binds to and modulates miR-128 function. Should this be the case, one would expect increases of the validated mRNA targets of miR-128, including the neurodegeneration and aging-related BMI1, SIRT1, and AXIN1 transcripts (Fig 6F; Chatoo *et al*, 2009; Lai *et al*, 2010; Min *et al*, 2013; Zhou *et al*, 2018). All of these miR-128 target transcripts were indeed upregulated in the SN of PD brains (*t*-test $P = 0.0004$ for BMI1, $P = 0.011$ for SIRT1, and $P = 0.005$ for AXIN1, Fig 6G). Moreover, our experimental work in PQ-exposed neuronal cell lines, where circSLC8A1 is upregulated identified similar increases in miR-128 targets (*t*-test $P = 0.03$, $P = 0.02$, and $P = 0.002$ for AXIN1, BM1, and SIRT1, respectively, Fig 6H). Nevertheless, under these conditions we did not detect changes in the levels of miR-128 itself (Appendix Fig S3G and H), indicating that the likely interaction of circSLC8A1 with miR-128 does not result in the degradation of this miRNA. To determine the relative concentrations of miR-128 and circSLC8A1, we performed a scaled qPCR from one of the SN samples and SH-SY cells (Dataset EV8). We indeed found that the molar ratio of circSLC8A1 and miR-128 is approximately 1:7 in both samples. As circSLC8A1 contains 7 binding sites for miR-128 in circSLC8A1, the resulting miRNA binding site/circRNA ratio is approximately 1 (0.73). This result is compatible with potential titration of the miRNA by the circRNA.

To further explore the functional relationship between miR-128 and circSLC8A1, we performed a miRNA enrichment analysis seeking all of the known validated targets of miRNAs which can predictably interact with our DE gene list from the SN of PD and control donors. Briefly, we crossed a list of all *homo sapiens* miRNAs and their validated targets with all of the expressed DE genes in the SN. This analysis revealed 18 miRNAs whose targets were enriched in our list of DE genes (Fig 6I), and miR-128 was among them (Fisher's exact test $P = 0.0036$, see Materials and Methods for details). Interestingly, the neurodegenerative disease-related and inflammation-suppressing miR-132 (Shaked *et al*, 2009) was also identified as a target-enriched miRNA (Fisher's exact test $P = 0.0332$). miR-132 is altered in several neurodegenerative diseases, including PD (Lau *et al*, 2013; Briggs *et al*, 2015) and AD (Lau *et al*, 2013), and it regulates genes related to molecular mechanisms of learning and memory, dopaminergic neuronal maturation, and protection from neuroinflammation and acute stress (Edbauer *et al*, 2010; Yang *et al*, 2012; Shaltiel *et al*, 2013). Taken together, these findings suggest that circSLC8A1 can bind and may modulate the activity of several miRNAs, including miR-128, and that without destroying miR-128, it can lead to miss-regulation of the PD-related miR-128 targets.

To further study the molecular mechanism of circSLC8A1, we performed a knockdown experiment in cell culture. For doing so, we designed and transfected shRNAs targeting the circSLC8A1 back-splicing junction into 293HEK cells. Indeed, transfection with the shRNA reduced the expression of circSLC8A1 by 70% compared to the siControl-treated cells (*t*-test $P = 0.0075$, Fig EV5A). Importantly, the shRNA did not affect the levels of the linear SLC8A1

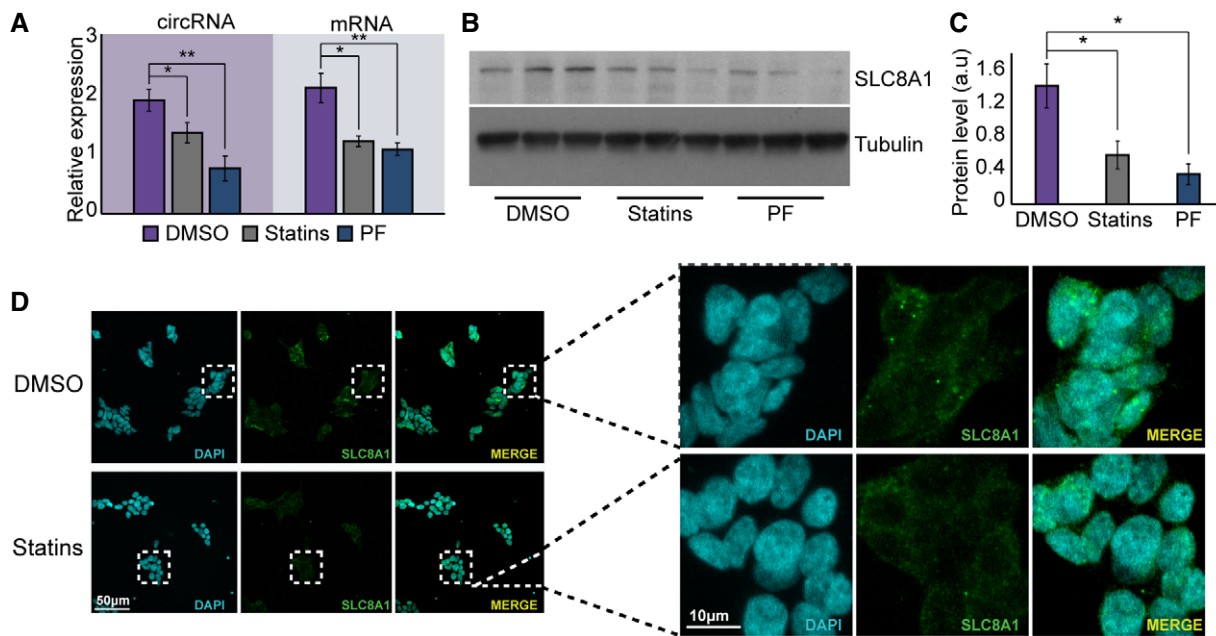

**Figure 5. CircSLC8A1 is reduced under neuroprotection.**

A    qPCR measurements of circSLC8A1 and SLC8A1 mRNA after SH-SY treatment of statins or the PF LRRK2 inhibitor, t-test **P = 0.007 and **P = 0.01 for PF-06447475 and *P = 0.047 and *P = 0.0136 for statins for cirRSLC8A1 and SLC8A1 mRNA, respectively, n = 3 biological replicas for each condition. Data presented as mean ± SD.

B, C    Protein gel and quantification of SLC8A1 and Tubulin as a loading control in statin and PF-treated neuronal cell cultures, t-test *P = 0.027 for statins and *P = 0.011 for PF-06447475, n = 3 biological replicas for each condition. Data presented as mean ± SD.

D    SLC8A1 immunostaining of statin-treated SH-SY cells.

mRNA (Fig EV5B). To determine the impact of the silencing of circSLC8A1 on gene expression, we performed RNA-seq from the control and circSLC8A1 KD cells. We identified 199 differentially expressed mRNAs (corrected $P < 0.05$, Supplementary information for full analysis), including 110 increased and 89 decreased genes (Fig EV5C, Dataset EV9). Enriched GO terms of those DE genes included RNA binding (corrected $P = 1.81E-4$), nucleoplasm (corrected $P = 1.43E-04$), SWI/SNF superfamily-type complex (corrected $P = 8.29E-03$), cytoplasmic stress granules (corrected $P = 5.56E-02$), and cell cycle process (corrected $P = 1.64E-02$, Fig EV5D). We have further evaluated predicted miR-128 targets. Indeed, 24 out of the 110 upregulated genes were predicted miR-128 targets (Dataset EV10), far more than expected by chance (Fisher's exact test statistic $P = 0.0433$). Together, this analysis has further strengthened the concept that circSLC8A1 might regulate miR-128 function, although it is not possible to determine if the effect is direct or indirect.

## Discussion

CircRNAs are highly expressed in the mammalian and human brain, and the levels of some of them are altered in Alzheimer's disease brains (Lukiw, 2013; Akhter, 2018). However, their importance for PD remained unknown. Our resource provides a valuable list of circRNAs expressed in the SN, AMG, and MTG of PD patients and healthy control brains, and our findings support the brain region specificity of circRNA abundance (Rybak-Wolf et al, 2015; Veno

et al, 2015). We further present efficient brain region-specific clustering of circRNAs. This was accompanied by DE coding genes in the PD brain regions, which primarily reflected known PD-affected pathways such as neurodegeneration and synaptic transmission (Miller et al, 2004; Zhang et al, 2005; Moran et al, 2006; Simunovic et al, 2009; Lewis & Cookson, 2012). Our RNA processing, library preparation, sequencing, and analysis pipeline could hence serve for exploring multiple PD-related issues. Intriguingly, healthy, but not diseased donor SN tissues showed modest age-accumulation of circRNAs, compatible with findings in several model organisms (Westholm et al, 2014; Gruner et al, 2016). We identified a decrease in circRNA abundance in the PD SN and focused on circSLC8A1 which was upregulated in the PD SN and strongly bound by AGO2 in neurons.

The SN is of special interest for PD studies, as this is where the loss of dopaminergic neurons leads to the characteristic symptoms of the disease (Farrer, 2006; Farrer et al, 2006). Interestingly, we found that this brain region presents some unique features regarding circRNA expression. First, we observed that the healthy SN presented considerably larger numbers of circRNAs compared to the MTG and AMG. As expected, due to the fact that circRNAs are generally enriched in neurons, individuals with PD showed lower numbers of circRNAs compared to healthy controls. Moreover, we observed an opposite trend in the AMG and MTG PD samples. Since RNA editing was previously found to be negatively correlated to circRNA abundance, we further measured global editing in our samples and in *Alu* elements within circRNA-producing exons and their surrounding introns. This analysis demonstrated significant

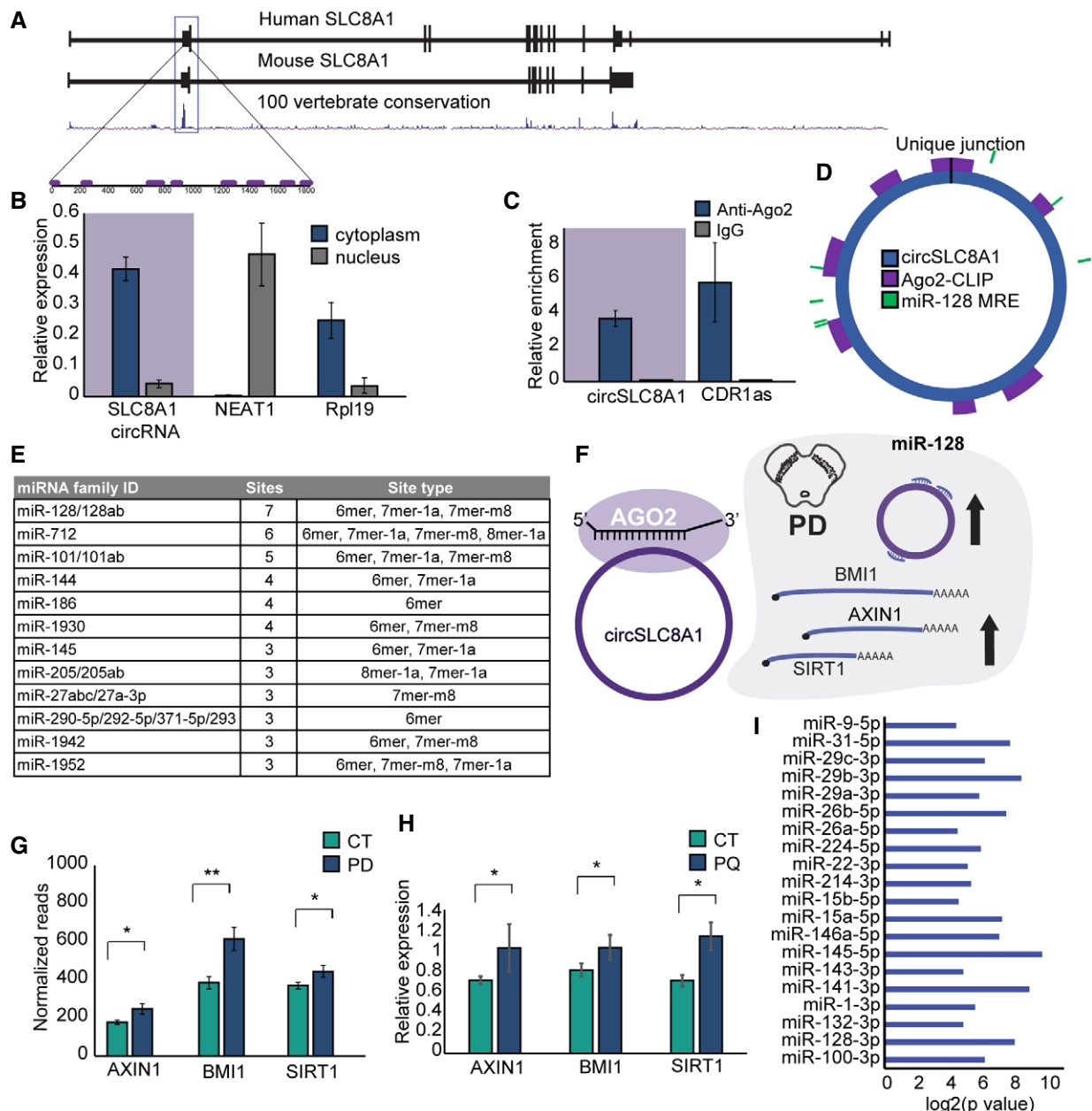

**Figure 6. Cytoplasmic circSLC8A1 binds Ago2 and regulates miR-128 targets.**

A   Gene structure and conservation of the human and murine SLC8A1 gene. The box marks the second exon that circularizes to create circSLC8A1.

B   qPCR quantification from nuclear/cytoplasmic fractionations of SH-SY cells, $t$-test $P = 0.005$ for circSLC8A1, $P = 0.009$ for NEAT1 and $P = 0.009$ for Rpl19. $n = 3$ biological replicas for each condition. Data presented as mean $\pm$ SD.

C   qPCR quantification of Ago2-bound RNA, immunoprecipitated (IP)/input values normalized to negative control for a known Ago2 target, circCDR1as, and circSLC8A1, $t$-test $P = 0.029$ for circSLC8A1 and $P = 0.001$ for circCDR1as, $n = 3$ biological replicas for each condition. Data presented as mean $\pm$ SD.

D   Schematic representation of circSLC8A1 with Ago2-binding sites from CLIP-seq and predicted miRNA binding sites.

E   miRNAs predicted to bind circSLC8A1 with 3 sites and more.

F   Schematic representation of circSLC8A1-Ago2 complex binding and its effect on miR-128 targets.

G   Relative expression of known miR-128 targets from SN RNA-seq, **$P = 0.0004$ for BMI1, *$P = 0.011$ for SIRT1 and *$P = 0.005$ for AXIN1.

H   Relative expression of known miR-128 targets in SH-SY cells treated with Paraquat, $t$-test *$P = 0.03$, *$P = 0.02$ and *$P = 0.002$ for AXIN1, BM1 and SIRT1, respectively. Data presented as mean $\pm$ SD with 4–6 biological replicas for each condition.

I   miR enrichment analysis of DE genes in SN from validated miR-targets.

decreases in global Alu editing in circRNAs and flanking introns in the PD's AMG and MTG. This trend anti-correlates with the higher levels of circRNAs in these regions in PD individuals as well as with previous reports of reduced editing activity in neurodegeneration (Singh, 2012; Lorenzini et al, 2018). Unexpectedly, the SN showed an overall lower editing compared to the other tissues and to control brains, possibly due to the lower levels of ADAR1 in this brain region. The lower editing rate in the SN correlated to higher circRNA abundance in comparison to other explored regions. Nevertheless, we did not observe a PD-related reduction in editing activity in the SN.

While in the healthy SN we detected age-related increases in circRNA levels, this correlation was lost in PD. Surprisingly, we could not find any age-related changes in other brain regions, either healthy or diseased. This might reflect insufficiently broad age ranges for identifying positive correlations (unlike other animals where such differences were found between developing, young, and aged individuals). Alternatively, or in addition, the lack of age-related changes in circRNAs production in the tested brain regions might reflect some uniqueness of the SN regarding circRNA expression and/or metabolism. For example, the unique SN pattern of circRNA profiles might reflect higher synthesis or lower circRNA degradation rates in the SN, which could also explain why the SN harbors more circRNAs overall. The circRNA profile in the SN further differed from that of mRNAs expressed in this brain region, with 26% of detected circRNAs but only 2% of mRNAs being SN-specific. This anomaly may indicate that the SN is much more prone for exon circularization, which could yield a far larger variety of circRNAs in this brain region compared to its mRNA repertoire.

SLC8A1 is a sodium–calcium exchanger responsible for one of the mechanisms in charge of excessive $Ca^{2+}$ removal from neuronal cell bodies (Khananshvili, 2014). As an efficient antiporter membrane protein, SLC8A1 removes calcium ions from cells after generating an action potential (He et al, 2009). This process is considered one of the most important cellular mechanisms for removing $Ca^{2+}$, which via a cytoplasmic process protects the mitochondrial space. Building high concentrations of $Ca^{2+}$ within the cell represents a well-established risk for initiating neurodegeneration, and dopaminergic neurons are especially susceptible to these insults (Wu & Kao, 2016). Most sodium/calcium exchangers similarly regulate calcium in dopaminergic and non-dopaminergic mesencephalic neurons. In comparison, dopaminergic neurons are thought to portray additional risk factors, including dopamine oxidation and heterogeneous expression of multiple types of sodium/calcium exchangers. This results in quantitative differences in calcium regulation and leads to exacerbated oxidative stress (Duda et al, 2016; Wu & Kao, 2016). Therefore, dopaminergic neurons may be under elevated risk of neurodegeneration, which correlates with a modified balance between circularization and canonical splicing that might change their survival ability under diverse insults. In this context, it will be very interesting to determine potential roles of the linear and circular forms of SLC8A1 RNA for seeking new approaches to deal with these insults.

Acutely stressed PQ-exposed cells showed dose-dependent elevation of circSLC8A1, decline of the SLC8A1 protein and increase in the levels of the miR-128 target transcripts, accompanied by exacerbated cell death (Berry et al, 2010). Inversely, the neuroprotective drug Simvastatin which is known for its antioxidant potential,

specifically in dopaminergic neurons, reduced both circSLC8A1 and the linear SLC8A1-derived protein. These seemingly conflicting results may indicate different circSLC8A1-related strategies of gene regulation, one causing elevation of circSLC8A1 at the expense of the linear transcript and the other reducing transcription from the SLC8A1 locus at large. In this context, it would be interesting to determine whether the levels of the SLC8A1 channel protein are affected in large cohorts of PD brains (particularly in the SN). While some of the detrimental effects of oxidative stress in vivo could be attenuated by the relatively larger half-life of protein channels compared to RNA transcripts, the complexity of the events we found reflects several altered checkpoints (circularized splicing, miRNA regulation, protein decline) which may amplify its impact.

Initial studies proposed miRNA sponging as the main molecular activity of circRNAs, but no conclusive evidence suggests such mechanism(s). For example, the circRNA CDR1as contains 71 binding sites for miR-7, yet knockout of this circRNA leads to decrease rather than increase in the levels of miR-7 (Piwecka et al, 2017). Our survey identified a putative link between circSLC8A1 levels and the function of miR-128 and its downstream target transcripts but sought links could not suggest any simple mechanism of action. Structurally, circSLC8A1 is classified as an AUG circRNA which spans over the canonical translation start site of the protein encoded from the host gene. Our results suggest that circSLC8A1 might regulate the levels and/or activity of miR-128. We do not have evidence regarding a direct interaction between the circRNA and miR-128, and the effect could be direct and/or indirect. Interestingly, other reports show binding of circSLC8A1 to Ago2. In cardiomyocytes, circSLC8A1 is bound to Ago2 but its specific miRNA regulation was not investigated (Siede et al, 2017). Our Ago2 CLIP data search supports the notion that circSLC8A1-Ago2 association in neurons accompanies functional binding to the circularizing exon that creates circSLC8A1. In this context, knockout mice lacking SLC8A1 cannot perform a spontaneous heartbeat and consequently die at early stages of development. In comparison, SLC8A1 is involved in NO-induced cellular toxicity in neuroblastoma cells, astrocytes, and microglia through a cGMP/protein kinase G (PKG)-dependent mechanism. This is accompanied by elevated $Ca^{2+}$ levels, ROS production, and phosphorylation of ERK, JNK, and p38 MAPK, which finally ends in apoptosis (Kitao et al, 2010). This circRNA also includes a potential Open Reading Frame which could encode for a short peptide that includes the N-terminal part of SLC8A1. However, a recent study failed to find such a peptide (Stagsted et al, 2019), leaving the miR-128 interaction as the main testable function of this circRNA. Nevertheless, future experiments suggesting a potential "sponging" activity of circSLC8A1 over miR-128 should take into consideration the stoichiometry of those molecules.

Despite these considerations, our data suggest a link between circSLC8A1 and miR-128. The exon forming circSLC8A1 contains 7 binding sites for miR-128, and Ago2-CLIP data likewise showed 7 binding sites for miR-128 on circSLC8A1. MiR-128 targets several mRNAs of highly relevant neurodegenerative and aging regulators, including silent information regulator transcript1 (SIRT1; Min et al, 2013). The activities of this deacetylase depend on and are regulated by nicotinamide adenine dinucleotide ($NAD^+$) and are thought of as protective and highly linked to aging processes, including regulation of protein homeostasis, neural plasticity, mitochondrial function, and sustained chronic inflammation (Min et al, 2013). miR-128 also

targets BMI1, a controller of free radical concentrations that affects neuronal survival and aging by repressing p53 pro-oxidant Activity (Chatoo et al, 2009). Moreover, miR-128 targets AXIN1, and over-expressed miR-128 protects dopaminergic neurons from apoptosis (Zhou et al, 2018). We found that these validated targets of miR-128 were all upregulated in PD brains, which might reflect the impact of miR-128 regulation by circSLC8A1 and its capacity to influence PD pathology and progression. However, this is only a correlation. In sum, we hypothesize that circSLC8A1 may operate to re-balance RNA processing while cooperating with other gene products to exacerbate or limit PD-induced damage as a neuroprotective agent.

In PD, disease symptoms may only appear when over 60% of dopaminergic neurons in the SN have died; and adjusting therapeutic protocols in treated patients is lengthy (Kalia & Lang, 2015). This calls for seeking therapeutic and prevention strategies to delay the onset and slow down the exacerbation of disease symptoms. CircRNAs are highly stable compared to other linear RNA molecules (mRNAs, miRNAs, or LncRNAs) and are therefore promising candidates for reflecting some of the disease-related changes in PD patients' neurons and serving as therapeutic targets (e.g., by identifying small molecule drugs suppressing their accumulation, such as Simvastatin). Notably, Simvastatin also operates to protect cultured neurons from oxidative stress via accelerating NEAT1 expression (Simchovitz et al, 2019) and LINC-PINT levels (Simchovitz et al, 2020). Alternatively, circRNAs may protect the diseased neurons from damage, in which case upregulating their levels would be commendable. In both cases, assessing neuronal response to tested therapeutics or preventive agents may add an important value to the basic research aspect of studying their role in PD's mechanisms of initiation and progress. Our current resource and accompanying experiments may be a step forward in that direction.

# Materials and Methods

### Human brain tissues, RNA isolation, and quality control

Postmortem brain tissues from PD and matched control donors were applied for and obtained from The Netherlands Brain Bank (NBB), Netherlands Institute for Neuroscience, Amsterdam. All tissues had been collected from donors for or from whom a written informed consent for a brain autopsy, and the use of the material and clinical information for research purposes had been obtained by the NBB. Clinical data for all donors are listed in Dataset EV1. Total RNA was prepared from frozen brain tissues using an mRNA Isolation Kit according to the manufacturer's protocol (Qiagen). The RNA was DNase-treated (DNaseI, NEB) and ethanol-precipitated. Sample quality control was assessed by Bioanalyzer (Agilent Genomics), and only samples which showed a RIN of above 6.5 were selected for sequencing. Other samples, with RIN below 6.5, were used for qPCR validations.

### Library preparations

Stranded ligation-based libraries were prepared using a modified protocol from (Shishkin et al, 2015) as follows: RNA was fragmented in FastAP Thermosensitive Alkaline Phosphatase buffer (Thermo Scientific) for 3 min at 94°C. Samples were cooled, and FastAP enzyme was added for 30 min at 37°C. Fragmented and dephosphorylated RNA was then cleaned up using 2.5× volume on SPRI beads (Agencourt) and then ligated to an internal sample-specific barcode (e.g., linker1 (5Phos/AXXXXXXXXXAGATCGGAA-GAGCGTCGTGTAG/3ddC/, XXXXXXXX) using T4 RNA ligase I (NEB). Ligated RNA samples were pooled into a single sample and cleaned using RNA Clean & Concentrator 5 columns (Zymo Research). Ribosomal RNA was removed from the pooled sample using the Ribo-Zero GOLD Kit (epicenter). RT was then performed for the pooled sample, with a specific primer (5′-CCTACAC-GACGCTCTTCC-3′) using AffinityScript Multiple Temperature cDNA Synthesis Kit (Agilent Technologies). RNA-DNA hybrids were degraded by incubating the RT mixture with 10% 1 M NaOH (e.g., 2–20 μl of RT mixture, 70°C, 12 min). The pH was then normalized by addition of corresponding amounts of 0.5 M AcOH (e.g., 4 μl for 22 μl of NaOH+RT mixture). The reaction mixture was cleaned up using Silane beads (Life Technologies), and second ligation was performed, where the 3′-end of the cDNA was ligated to linker2 (5Phos/AGATCGGAAGAGCACACGTCTG/3ddC/) using T4 RNA ligase I. The sequences of linker1 and linker2 are partially complementary to the standard Illumina read1 and read2/barcode adapters, respectively. Reaction mixture was cleaned up (Silane beads), and PCR enrichment was set up using enrichment primers and Phusion HF Master Mix (NEB). To minimize amplification-induced errors, only 10 cycles of enrichment were performed. After clean-up with 0.8× volume of SPRI beads, libraries were ready for characterization by TapeStation Instrument—Agilent Genomics. Remaining ribosomal RNA levels were 8–12%.

### Cell culture

Human neuroblastoma (SH-SY5Y) cells were grown at 37°C and 5% CO2 in a 1:1 mixture of EMEM and Ham's F12 medium (Gibco BRL) with 2 mM L-glutamine (Gibco BRL), 10% fetal bovine serum (FBS). Differentiation of these cells was carried out by adding 10 mM of retinoic acid (RA) twenty-four hours after plating the cells. After 5 days in the presence of RA, cells were subjected to paraquat (PQ) exposure by incubation with different concentrations of PQ (Sigma) in complete media for 24 h at 37°C, with or without neuroprotective agents.

SH-SY5Y cell line was purchased from ATCC in 2017 (CRL2266), HEK293 cells, passage 2 received from the Meshorer laboratory that purchased it from ATCC 2014 (CRL-1573).

### siRNA treatment

Protected siRNA molecules targeting the unique junction of circSLC8A1 (IDT) were transfected in 6-well plates using hiPerfect transfection reagent (Qiagen) according to the manufacturer's instructions:

circSLC8A1:

Sequence 1 mUmGrA mArAmU rUmGrU mUrAmG rGmUrU mGrUmG rA

Sequence 2 rUrCrA rCrArA rCrCrU rArArC rArArU rUrUrC mAmUrU

Negative control:

Sequence 1 mCmGrU mUrAmA rUmCrG mCrGmU rAmUrA mArUmA rC

Sequence 2 rGrCrG rUrArU rUrArU rArCrG rCrGrA rUrUrA mAmCrG

## hiPSC cell differentiation

hiPSC colonies were induced for neuronal lineage differentiation using dual SMAD inhibition by blocking the two signaling pathways that utilize SMADs for transduction: BMP and TGFβ. hiPSC colonies were separated and seeded on Matrigel as single cells and incubated with human induction media (DMEM/F12, Neurobasal, N2, B27) supplemented with 2 SMADi (20 µM SB, 100 nM LDN) and human recombinant DKK1 (or its equivalent XAV-939) for 10 days. Then, the cells were incubated with neural induction media supplemented with SHH and DKK for additional 10 days, inducing the cells to become neural progenitor cells (NPCs), expressing PAX6, Sox2, and Nestin.

## Computational procedures

To detect circRNA and mRNA transcripts, we constructed RNA-seq libraries by using the rRNA depletion method, enabling simultaneous detection and profiling by sequencing of both circular RNA forms and linear mRNA transcripts in these samples (Fig EV1C and D). All sequences were also analyzed for mRNA expression by the use of bowtie2 and STAR in order to accurately align them to the transcriptome. In the pooled libraries, rRNA abundance ranged between 8–12% and the alignment rate of the remaining transcripts to the genome (w/o rRNA) ranged between 76–82%. We then used a dedicated bioinformatics pipeline to detect and annotate circRNAs (Memczak *et al*, 2013; Pamudurti *et al*, 2017). Reads supporting particular head-to-tail junctions were used as an absolute measure of circRNAs abundance and were normalized using the DESeq2 algorithm in R. This analysis identified thousands of circRNAs, with part of those differentially expressed between PD patients and control volunteers. For DESeq2 normalization, we normalized the numbers of total circRNAs detected by adding all mapped reads from the STAR alignment (mRNAs, lncRNAs, etc.) according to the aligned reads which were detected and quantified in each sample. Additionally, we used the Sailfish pipeline to achieve alignment-free isoform quantification from RNA-seq reads using lightweight algorithms (Langmead & Salzberg, 2012; Kim *et al*, 2013; Patro *et al*, 2014)

The resultant libraries were sequenced deeply (~50 M reads per sample on average), allowing reliable detection and quantification of mRNAs, non-coding RNAs and especially circRNAs. The large numbers of sequenced samples further assisted in dealing with the individual heterogeneity characteristic of human samples and especially of diseased brain tissues. Specifically, mRNA reads were normalized, and differentially expressed transcripts were analyzed using the DEseq2 algorithms in R. Dataset EV2 details the numbers of reads in each pool of samples and of circRNAs detected within each pool of samples, and the number of reads in each sample. Generally, similar numbers of circRNAs were detected (roughly 12K circRNAs among 50 M sequenced reads in each sample). We excluded all samples with < 5,000 detected circRNA reads in the analysis, as detailed in Fig EV1E–G, excluded samples included MTG2, MTG10, SN1, SN16, and SN19. A total of 38,860 transcripts could be quantified: 36,311 mRNAs and 2,549 circRNAs. For quantification of circRNA abundance in each tissue and condition, circRNAs were included in the analysis only if they were expressed

> 5 or > 10 reads per tissue. For differential expression and GO analysis, we corrected for cell composition according to three cell markers, including transmembrane protein 119 (TMEM119) for microglia, aldehyde dehydrogenase 1 family member L1 (ALDH1L1) for astrocytes, and synaptotagmin 1 (Syt1) for neurons using EdgeR package in R (Robinson *et al*, 2010; McCarthy *et al*, 2012). Enriched GO terms were analyzed using DAVID Bioinformatics Resources (da Huang *et al*, 2009).

Sequencing samples were only analyzed if number of circRNA reads were > 5,000. RNA quality was pre-established. Preparation of RNA libraries was performed with samples unmarked as PD or control. RNA sequencing data were checked for variance and normalized by the use of EdgeR and DEseq2 packages in R.

## Detailed GO term analysis of WGCNA modules correlated with disease, tissue, and sex

Enriched GO terms that indicate pre-were associated with the brown module include poly(A) RNA binding (GO:0044822, $P$ = 1.77E-07), nucleic acid binding (GO:0003676, $P$ = 1.87E-07), mRNA splicing, via spliceosome (GO:0000398, $P$ = 2.15E-07) and regulation of transcription, DNA-templated (GO:0006355, $P$ = 5.57E-07). The red module was associated to GO terms such as response to unfolded protein (GO:0006986, $P$ = 6.54E-04), poly(A) RNA binding (GO:0044822, $P$ = 7.69E-04), mitochondrial matrix (GO:0005759, $P$ = 0.003841), and post-transcriptional regulation of gene expression (GO:0010608, $P$ = 0.007285). The cyan module that was associated with the sample gender included many gender-related genes, such as DDX3Y (DEAD-box helicase 3, Y-linked), XIST (X inactive-specific transcript), NLGN4Y (neuroligin 4, Y-linked), and others.

In addition, the blue module which was significantly correlated with the tissue parameter included various neuronal functionality GO terms, such as cell junction (GO:0030054, $P$ = 8.12E-15), post-synaptic density (GO:0014069, $P$ = 1.81E-12), voltage-gated potassium channel complex (GO:0008076, $P$ = 1.79E-11), dendrite (GO:0030425, 2.57E-10), and synaptic vesicle membrane (GO:0030672, $P$ = 2.46E-05). These GO terms indicate pre- and postsynaptic regulation differences in the different tissues and also include specific neurotransmitters regulation: glutamate secretion ($P$ = 9.16E-07), cholinergic synapse ($P$ = 1.41E-06), dopaminergic synapse ($P$ = 6.35E-06), and GABAergic synapse ($P$ = 8.01E-06).

## RNA Editing analysis

Raw FASTQ quality was assessed using FastQC (https://www.bioinformatics.babraham.ac.uk/projects/fastqc/), and PCR duplicates were removed with PRINSEQ (http://prinseq.sourceforge.net/). Remaining reads were uniquely aligned to a human reference genome (hg38) using STAR (Dobin *et al*, 2013; version 2.6.0c) with minimum overhang for spliced alignments of 8 and maximum intron size of 1,000,000 (ENCODE standard options). *Alu* elements in circRNA sequences and flanking regions were obtained using BEDTools (Quinlan & Hall, 2010, versions 2.26.0 and 2.27.1) and curated UCSC RefSeq annotations. *Alu* editing index was computed by the previously presented approach (Roth *et al*, 2019). Briefly, it measures the averaged editing level across all *Alu* adenosines, weighted by their expression. This index averages over millions of adenosines and is, therefore, rather robust to statistical noise (Paz-

**The paper explained**

**Problem**

Parkinson's disease (PD) is the leading neurodegenerative movement disorder; however, the molecular mechanisms underlying cellular degeneration in PD brains remain poorly understood. Circular RNAs (circRNAs) are a recently discovered class of RNAs but their roles and expression patterns in the healthy and diseased human brain are largely unknown.

**Results**

We used very deep RNA sequencing to explore the expression patterns of circRNAs in three different brain regions from PD patients, including the substantia nigra (SN) where dopaminergic neurons die in the diseased brain, the amygdala which responds to stressful situations at large and the temporal gyrus harboring the routes from deep brain nuclei to the cortex. We found CircRNA levels to be brain region-specific and inversely correlated to RNA editing. Also, we identified age-dependent accumulation of circRNAs in the healthy SN but not in the PD SN, where this correlation is lost and the total number of circRNAs is reduced. We focused our experiments on CircSLC8A1, which originates from the Ca$^{2+}$ regulating SLC8A1 gene. We found CircSLC8A1 increases in the SN of PD individuals and in cultured cells exposed to the oxidative stress-inducing agent Paraquat. Notably, CircSLC8A1 carries 7 binding sites for one microRNA, miR-128, and is strongly bound to Ago2. Correspondingly, we found increases in RNA targets of miR-128 in PD brains, suggesting that circSLC8A1 regulates the function and/or activity of miR-128.

**Impact**

Our findings establish a resource of circRNAs expressed in the brain of PD patients, reveal previously unknown links between circRNA expression, oxidative stress and PD pathology, and call for exploring the implications of circSLC8A1 accumulation for addressing the initiation of PD neurodegenerative processes.

Yaacov *et al*, 2015). Analysis and statistics were conducted using R (R Foundation for Statistical Computing; https://www.R-project.org/). Samples were filtered to match the samples used for circRNA expression analysis by the expression thresholds applied.

## Cell-type enrichment calculation

Cell-type gene expression was found in (Zhang *et al*, 2014a,b). Specific genes were considered as expressed in a specific cell type if the expression value from the uncorrected RNA-seq data for cell composition has an order-of magnitude higher than the average expression level of all cell types plus 5× standard deviation.

## miR enrichment analysis

A list containing all manually curated human miRNA and target sites was retrieved from miRTarBase, and Fisher's exact test was used to determine significant enrichment for miR-128 targets among DE genes in PD SN. Diana prediction tool was used for target prediction (http://diana.imis.athena-innovation.gr/DianaTools/index.php).

## Stoichiometry measurement

To perform the Stoichiometry measurement, we amplified an amplicon that contains the circRNA qPCR product from 293 HEK cells and cleaned the amplicon to use as a synthetic template. For miR-128,

we ordered a short ssRNA molecule of miR-128 exact sequence: UCACAGUGAACCGGUCUCUUU. We used these templates for generating a standard curve which we used to quantify the amount of circSLC8A1 and miR-128 in SH-SY and in one of the SN samples. The primer sequences we used to generate the circSLC8A1 qPCR template were as follows: F - CACGCAGGCATTTTTACTT, R – TACATGGTCCACATGGGAAA, full calculations and calibration curve are present in Dataset EV8.

## qPCR

cDNA was generated using iScript-select kit (Bio-Rad) and was utilized as a template for quantitative real-time PCR performed with the C1000 Thermal Cycler Bio-Rad. The PCR mixture contained Taq polymerase (SYBR green Bio-Rad). miRNAs were quantified using TaqMan miRNA Assays (Thermo Fisher).

## qPCR primers

| SLC8A1 circRNA/mRNA F | GGTGGAGGGGAGGATTTTGA |
|---|---|
| SLC8A1 circRNA R | ACTTAATCGCCGCATGTTGT |
| SLC8A1 mRNA R | TCCAATCTCAAGGAAGAAGGTCT |
| Actin F | CCCAAGGCCAACCGCGAGAA |
| Actin R | AGTGGTACGGCCAGAGGCGT |
| TUBB3 F | GCAACTACGTGGGCGACT |
| TUBB3 R | GGCCTGAAGAGATGTCCAAA |
| AXIN1 F | CGCGGGACAGATTGATTCAC |
| AXIN1 R | CAGTTCTCCCTCCTCACCAG |
| BMI1 F | AGATACTTACGATGCCCAGC |
| BMI1 R | GGTCGAACTCTGTATTTCAATGG |
| SIRT1 F | AGCAGATTAGTAGGCGGCTT |
| SIRT1 R | GACTCTGGCATGTCCCACTA |

## Western blot

Proteins were extracted using RIPA buffer supplemented with protease inhibitors (cOmplete™, EDTA-free Protease Inhibitor Cocktail). Equal amounts of total protein from all fractions were resolved on 4–12% polyacrylamide gels (Criterion gels, Bio-Rad), blotted onto membranes, and probed with antibodies against SLC8A1 (ab177952, 1:1,000) and Tubulin (1:1,000, Abcam).

## IHC

SH-SY cells grown on coverslips coated with Poly-L lysine were fixed with 4% PFA (Electron Microscopy Sciences) for 15 min and then washed in D-PBS. Cells were permeabilized with D-PBS/0.2% Triton X-100 for 1 h, then blocked in 3% Normal Donkey serum in D-PBS/0.2% Triton X-100 for 1 h. Next, cells were incubated O.N. at 4c with 1:100 dilution of anti-NCX1 antibody (Abcam—ab2869) in blocking solution. Cells were washed in D-PBS/0.2% Triton X-100 and incubated for 1 h at RT with 1:500 dilution of goat anti-mouse Cy3-conjugated secondary antibody (Jackson immunoresearch 115-166-072). Finally, cells were washed in D-PBS, stained with DAPI for 10 min, washed, and mounted on microscope slides.

## Confocal Imaging and image analysis

Slides were scanned using a confocal laser scanning microscope (Olympus, FluoView FV10i) in a 60× water lens. Briefly, 10 randomly selected regions of interest (ROIs) from each slide were imaged taking a z-stack of 20 μm depth to cover the entire area of the cells. Images were z-projected (maximum intensity), and the DAPI signal was morphologically detected to automatically estimate the region of the cells in each image. Next, All DAPI-selected regions were measured for mean fluorescent intensity in the Cy3 channel (FIJI). Independent sample two-tailed *t*-test statistical analysis was performed using Microsoft Excel.

## Tissue retrieval statement

Informed consent was obtained from all subjects or their care takers to ensure that the experiment conformed to the principles set out in the WMA Declaration of Helsinki and the Department of Health and Human Services Belmont Report. At the Hebrew University, performing of this study had been approved by the Committee for Studies Involving human-originated materials and tissues.

# Data availability

The datasets produced in this study are available in the following databases: RNA-seq data: Gene Expression Omnibus GSE133101 (https://www.ncbi.nlm.nih.gov/geo/query/acc.cgi?acc = GSE133101).

**Expanded View** for this article is available online.

## Acknowledgements
The authors acknowledge support by the Michael J. Fox Foundation for Parkinson's Research Grant ID 11183 (to SK and HS), the National Institutes of Health, Grant R01AG057700 (to SK), The Netherlands Brain Bank (NBB) for brain tissues (to HS), the Israeli Ministry of Science, Technology and Space, Grant No. 53140 (to HS), the Edmond and Lily Safra Center of Brain Sciences (ELSC) for MH's post-doctoral fellowship and the Clore Foundation for AS's pre-doctoral fellowship.

## Author contributions
Scientific guidance: HS and SK; Planning and experiment: MH; Purification of RNA from human brains: AS; SH-SY: Experimental analysis with NM, SV, and MH; IHC experiment and microscopy: NY; RNA Editing analysis: MK, RC-F, and EYL; ESC experiments: TMR, MM, and EM; Manuscript writing: MH, SK, and HS; Manuscript editing: ERB and DSG; and all co-authors read and approved the final version.

## Conflict of interest
The authors declare that they have no conflict of interest.

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
