## [Review Process File · EMBO Molecular Medicine]

A Parkinson's disease CircRNAs Resource reveals a link between circSLC8A1 and oxidative stress

Mor Hanan, Alon Simchovitz, Nadav Yayon, Shani Vaknine, Roni Cohen-Fultheim, Miriam Karmon, Nimrod Madrer, Talia Miriam Rohrllich, Moria Maman, Estelle R. Bennett, David S. Greenberg, Eran Meshorer, Erez Y. Levanon, Hermona Soreq, and Sebastian Kadener

DOI: [10.15252/emmm.201911942](https://doi.org/10.15252/emmm.201911942)

Corresponding authors: Sebastian Kadener (skadener@brandeis.edu), Hermona Soreq (hermona.soreq@mail.huji.ac.il)

Review Timeline:

Submission Date:	13th Jan 20
Editorial Decision:	21st Feb 20
Revision Received:	8th May 20
Editorial Decision:	28th May 20
Revision Received:	23rd Jun 20
Accepted:	25th Jun 20

Editor: Celine Carret

Transaction Report:

21st Feb 2020

Dear Prof. Kadener,

Thank you for the submission of your manuscript to EMBO Molecular Medicine. We have now heard back from the three referees whom we asked to evaluate your manuscript.

You will see that the three referees find the study to be an interesting and valuable resource. However, they recommend further efforts on figure presentation, provision of statistical values, tests description, and modification of the main text (legends have to be expanded, clarifications and explanations are needed, refocus of the main findings, development of some of the results, tone down of causality claims, provide genesets, a new title...). In terms of experimental work, neuronal loss should be analysed (highlighted by 2 referees), circSLC8A1 and miR-128 ratio evaluated, oxidative stress effect on global transcription looked at. In terms of analysis, the cell type data should be deconvoluted and miR-128 targets analysed globally.

We would therefore welcome the submission of a revised version within three months for further consideration and would like to encourage you to address all the criticisms raised as suggested to improve conclusiveness and clarity. Please note that EMBO Molecular Medicine strongly supports a single round of revision and that, as acceptance or rejection of the manuscript will depend on another round of review, your responses should be as complete as possible.

I look forward to receiving your revised manuscript.

Yours sincerely,

Celine Carret

Celine Carret, PhD
Senior Editor
EMBO Molecular Medicine

*Additional important information regarding figures and illustrations can be found at <http://bit.ly/EMBOPressFigurePreparationGuideline>

***** Reviewer's comments *****

Referee #1 (Remarks for Author):

The manuscript 'A CircRNAs Resource indicates an AGO2-related regulatory role for circSLC8A1 in the Parkinsonian brain' represents a thorough attempt to identify circRNAs that are functionally related to the pathophysiology of Parkinson's disease. Notably, the authors make use of a high-quality collection of postmortem human brain samples, namely the one of the Netherlands Brain Bank in order to draw disease-relevant conclusions for the human PD brain. CircRNAs are emerging as a ncRNA species with a wide array of regulatory roles in the healthy and diseased CNS. Nevertheless, the understanding of their cross-interactions with the degenerating brain is currently limited, and studies like the one reported here can provide a valuable resource of information that could be then used for independent validation. However, the manuscript in its current form, lacks several pivotal elements of appropriate analysis and interpretation, as discussed below.

Major remarks:

- One of my greatest concerns is the fact that the authors try to interpret data derived from bulk tissue RNAseq, without acknowledging the limitations of such an approach and the confounding variables that it may introduce into the analysis of the data. Ideally, cell type deconvolution tools should be used to address any issues of this sort.
- Related to the previous point, how can the authors know whether the lower circRNA levels in PD SN are not caused by neuronal loss? Instead, the authors attribute this observation to a particular PD-specific regulatory mechanism. This should be addressed.
- Alu editing index does not significantly differ between CTR and PD SN. This suggests that RNA editing is not mechanistically relevant to PD pathology. The authors should discuss. How could then the differential expression of circSLC8A1 in PD SN (compared to CTR) be explained?
- No causal relationship between circSLC8A1 and miR-128 can be supported by the current data. The authors should either tone down the pertinent passage or address this issue by assessing the levels of miR-128 in vitro following transfection of circSLC8A1.
- The statistical tools used for each experiment should be clearly indicated. What type of statistics was used in Figure 4D?

Other remarks:

- The authors should provide the genesets used for pathway analysis.
- The levels of all microRNAs indicated in Figure 6E should be assessed.
- It is not entirely clear how only samples with RIN values above 6.5 were used for library preparation (as indicated in the text), while in the pertinent Sup. Table1, 42 samples do not fulfill those criteria.
- The direction of pathway enrichment (repressed/induced) should be indicated for all GO term enrichment analyses.
- The terms 'upregulation' and 'downregulation' should be used with caution taken that any apparent gene level changes may be driven by differences in cell numbers in the bulk analysis.

- The authors should provide average values (and corresponding statistics) for the within region variability of their CTR vs PD cohorts (analyzing differences of gender, age, Braak stage, Amyloid stage, PMI).
- PMI values should be provided for all samples.
- Fold change, p-value and corrected p-value for all identified circRNAs should be included.
- There are several typos.

Referee #2 (Remarks for Author):

Hanan and colleagues performed RNA-seq to identify changes in circular RNA expression in the brains from controls and donors with Parkinson's disease. Overall trends are described, including suggestions about circRNA biogenesis via changes in Alu editing levels. The authors then focus on circSLC8A1 which is upregulated in the PD SN. A model is proposed in which circSLC8A1 binds miR-128 and functions as a sponge. The manuscript is dense but generally clear and I suggest a few additional analyses.

Major point:

(1) My main concern is about stoichiometry. What is the relative ratio of circSLC8A1 to miR-128 in cells? They should be at similar levels in order for the circRNA to function as a sponge. A related point that the authors should address: what is the relative amount of linear vs circular RNA generated from the SLC8A1 gene?

(2) Fig 6G/H: The authors show 3 miR-128 targets that increase in PD brains, but these are cherry picked mRNA targets. What percentage of miR-128 targets increase in PD brains? If the authors' model is correct, most direct miR-128 targets should show increased expressed levels.

Minor points:

(1) Last sentence of abstract is over-sold. The current manuscript does not provide experiments that suggest a preventive value for modulation of circSLC8A1. If the authors want to make such a claim, they need to show changes in cell survival etc when circSLC8A1 expression is experimentally modulated.

(2) P.2: "However, it remained unclear what are the consequences of this binding (Piwecka et al)" is an odd phrase because the CDR1as mouse model described in the cited paper does suggest a mechanism.

(3) P.3: The authors may also wish to mention that the presence of multiple Alu elements in introns can allow alternative circularization events.

(4) Supplementary Fig 2G should have error bars to show the variation in expression levels across samples.

(5) Page 7: When describing the circRNA profiling, it would be helpful to more clearly clarify the criteria used. What is a circRNA expressed at "very low levels"? How many sequencing reads must be observed for a circRNA to be annotated? Do the circRNAs need to be detected by both algorithms? What was the degree of overlap in circRNA predictions between algorithms? Some of

these points are addressed in the methods but they would be helpful to also have in the main text.

(6) The callouts for Supplementary Fig 3B and 3C are reversed on page 7.

(7) Fig 2A: I find the model to be confusing as drawn. Back-splicing does not lead to a hairpin shaped transcript. The hairpin shaped transcript occurs first and then back-splicing occurs.

(8) Fig 3C: The authors should comment on why they think editing levels decrease in the SN and why reduced levels of circRNAs are observed. One would have expected increased circRNA levels as is seen in the MTG and AMG.

(9) Page 9: "Surprisingly, we did not observe positive correlation in any of the other two assayed tissues" - Please show these data.

(10) Figure 4A could be more informative, e.g. what exon(s) are included in the circRNA? Is it flanked by Alu repeats? A related question: is editing observed at the SLC8A1 locus?

(11) Figure 4D: Please provide statistics for mRNA between CT and PD. What is the measured p-value?

(12) Figure 4E: Please provide statistics.

(13) Page 13: When referring to the published Ago2-CLIP data, please make clear that one cannot distinguish binding to linear vs circRNA unless the CLIP reads span the backsplicing junction.

Referee #3 (Remarks for Author):

The report of Hanan et al. entitled CircRNAs Resource indicates an AGO2-related regulatory role for circSLC8A1 in the Parkinsonian brain describes for the first time a catalog of CircRNAs (together with mRNAs and microRNAs) present in three brain regions derived from Parkinson disease patients and healthy controls and points towards circSLC8A1 as an Ago2-associated circRNA, regulated by oxidative stress, and abundant in Parkinson Disease (PD)-derived substantia nigra that influences mir-128 activity.

Major findings/novelty of the manuscript:

This is the first large and systematic study linking circRNAs to PD. For doing so the authors applied a tour de force approach and sequence 3 brain regions in dozens of control and PD individuals. They find that circRNAs globally change in the brain of PD individuals (in all the 3 regions studied but with opposite results between the SN and the other 2 regions). This is a key and relevant finding and surprisingly it is not highlighted enough in the manuscript (not even in the abstract). Moreover, the authors establish a link between circRNAs and PD by finding global changes of circular transcripts in different brain regions. They describe a differentially expressed set of circRNAs in the substantia nigra (SN) of PD individuals. Additional relevant findings are i- the pronounced brain region-specificity of gene expression patterns, ii- and the age-dependent circRNA accumulation in the human brain.

Finally, authors focus on one highly expressed CircRNA in PD material and perform an initial molecular characterization.

This is certainly a timely and relevant resource paper. It is based on a dataset generated directly from a large and well-curated set of human brain material derived from PD patients and healthy controls. Additionally, the massive sequencing on which the Kadener's lab has an extended experience and the different ways to present and analyze the data add additional value to the work. Having said that, there are numerous points which, though they do not put under question the main findings of the manuscript might certainly contribute, at least in the eyes of this reviewer, to a more clear presentation and interpretation of the data.

Main Comments

1. The writing is in many passages too convoluted and difficult to follow. The text will greatly benefit from conceptual and language simplification. In addition, the authors tend to overstate their findings, which is not necessary and sometimes deviates the attention of the reader. For example: There is no clear indication that circSLC8A1 has a causative role on PD so, it is not necessary in the abstract to indicate that the results "advocate a preventive value for its modulation".

2. Another issue of relevance is related to the main and running titles of the manuscript. The main title suggests a causal ("regulatory") role of AGO2-related regulation of circSLC8A1 in PD and the running title states advocates for a "modulatory" effects of circSLC8A1 via Ago2. This is mainly a resource paper and these titles sound close to overstatements since a causal role of the circular form of SLC8A1 is actually not proven in the manuscript. Taking this into account I would suggest a title that more strictly described the main findings of the study.

3. The authors should better describe why they selected MTG and AMG as additional brain areas. Actually, both brain structures are involved in the control and implementation of stress-related programs and anxiety-like behaviors what makes them also particularly interesting per se, but also in the context of PD since anxiety is also an often associated symptom in PD patients. Along this line, and considering that the present study is intended to be a resource article, the authors should invest more efforts to describe in more detail the catalogue of mRNA and circRNAs identified not only in the SN but also in MTG and AMG brain regions both in PD and control samples. For instance, which are the circRNAs present in the amygdala of healthy and PD individuals? I don't find this kind of data clearly described in the paper, neither in figures nor in tables and I have the feeling the paper would greatly benefit from precisely described and list the candidates from these brain areas.

4.a) Regarding the presentation of the sequencing data: almost all data in the paper seems to be focused on up-regulated candidates but downregulation is equally relevant and should also be clearly described both in text and figures. The term differentially expressed (DE) genes is not accurate enough. For instance, the data presented in supplementary table 3 might be organized dividing up-regulated and down-regulated and not merely by FDR. Moreover, the co-regulation data presented in Fig 1H-J is constructed from up-regulated genes? what about the existence of down-regulated modules?

b) And regarding the co-regulated modules, I don't find which are the genes that constitute the modules. If I am right, the authors should describe them in a supplementary figure; at least the most significantly different modules.

c) In the particular case of circRNAs, it would be interesting if the authors could add a major level of detail to the presented data, for instance, the distribution of sizes, whether the circRNAs are intronic vs exonic, which types of regions they contain (5' UTR, ORF, 3' UTR), number of exons or exact junctions, etc.

5. In page 7 the authors described that "as much as 26% of the detected circRNAs but only 2% of the mRNAs were unique to the SN (Figure 2E, grey), whereas 19% of the identified circRNAs but 82% of the mRNAs were shared between all tissues (Figure 2E, green)."

This is very important finding. It should be underlined and the authors should elaborate more about it in the discussion.

6. In page 7 second paragraph the authors state that "...the SN expressed higher total numbers of circRNAs compared to the MTG and the AMG (normalized to library total reads, Figure 3A,"

This is very interesting finding that raise the question as to whether the SN also express higher levels of mRNAs transcripts too. Consequently, how about the comparison of the expression levels of their linear counterparts? Do they inversely correlate?

7. In Fig2C, the data described includes both healthy controls and PD? Again, the authors should rather show the data about differentially expressed RNA species in each separate brain regions comparing healthy and PD patients.

8. The data presented suggest that the circSLC8A1 somehow regulates miR128 but it is not clear how (and it is likely beyond the scope of this study), so the authors should limit to state with caution how the circRNA might modulate the miRNA.

9. As circSLC8A1 seems to be upregulated in PD and by oxidative stress, it would be interesting to know whether oxidative stress globally induces transcription from the locus or induces circularization of the exons. This could be easily done in cell culture by qPCR from intronic sequences upon exposure to PQ. Moreover, the authors should look for inverted repeats or potential RBP binding sites. Any finding (even if no sequences are found) would be informative.

Minor points

A. If the submission format allows it, in case a putative new version of the manuscript should be reevaluated, I would thank the authors the addition of the following features to the submitted document:

Page numbers

Line numbers

Accommodate figures and their corresponding legends together

B. The legends are often so concise that basic information to properly understand the figures is missing.

C. In page 7 last sentence authors write: "This suggested a particular regulation and maybe importance of circRNAs in the PD SN"

This reviewer wonders whether the differences could be at least partially due to neuronal loss. The authors should mention and discuss this possibility.

D. In page 9 one can find the following sentence "We conclude that RNA editing and RNA circularization are anti-correlated in PD and control brains." I am confused with this statement because this anticorrelation seems to be lost in the SN because the editing does not change in that area. Please clarify.

E. In page 9 second paragraph the authors claim the following regarding the age-dependent

accumulation of circRNAs "Surprisingly, we did not observe positive correlation in any of the other two assayed tissues, but this could be due to the limited age range of the assayed samples". I don't find it described in figures. Is this a "not shown" data? In my view, it should be shown.

F. In page 9 the third paragraph starts as follow: "We then look for differentially expressed (DE) circRNAs in the brain tissues from PD and healthy individuals. We indeed identified 24 DE circRNAs (corrected p value < 0.05) between control and PD tissues (Figure 3H)." Is this a pooled data emanating from all brain regions together or just from SN? Please clarify and in case the first case, please explain why.

G. In Fig 3H what are the red dots? Significant DE candidates? This is a relevant figure. It might be important to clarify this and provide a clear-cut table with those significant (and specially relevant) circRNAs. In addition the short names of those genes might also be written in the figure, perhaps making the figure a bit larger.

H. In page 13 at the end of first paragraph the following sentence "However, the levels of this circRNA remained unchanged in control and PD fibroblast samples (Schulze, Sommer et al., 2018) (Supplementary Figure 6C). This might indicate that under normal growth conditions, the regulation over circularization and therefore the balance between circSLC8A1/SLC8A1 expression does not change, even in the case of genetic PD background." These results are confusing to me. I would recommend reformulating the sentence.

I. Immediately after, one can reads "The latter result suggests that changes in circSL8A1 expression in the PD brains might be related to other aspects of PD like oxidative stress." I am not clear which "other aspects" the authors really mean since the previous sentence is referring to differentiation process of fibroblast as starting material, which are not strict "aspects of PD". The sentence is unclear to me. Perhaps the authors mean that the changes in circSLC8A1 might be secondary to cellular insults or challenges reported to occur in dopaminergic neurons such as oxidative stress. If so, this should be formulated in a more clear way.

J. In the Fig 4I only the blots are shown. A quantification of the blots would be desirable.

K. In data described in supplementary figure 6C is somehow difficult to follow since the legend provides almost no details. Which kind of sample each column represents? Is this graph showing previously published data mixed with original data form the paper? Unclear to me, but if so, that should be clearly described and stated. On the other hand, the title of Supplementary Fig 6 mentions the use of ES cells, not iPS Cells, which kind of cells have been actually employed in this figure?

L. Many citations are wrongly described. Multiple times only the start (but not the end) page is written (e.g. Gal-Mark et al, Grunner et al, Holdt et al., Langfelder et al, Min et al., etc), In many other cases no pages are described at all (e.g. Agarwall et al or Piwecka et al.). On the other hand there are no spaces between citations what makes difficult the rapid finding of the citations. Please correct these mistakes.

M. In page 7 the mention of figure 3B and 3c seem to be interchanged.

N. Some graphs with Cartesian axes have major ticks but most of others graphs do not have. The ticks help the reader to more precisely evaluate the graphs; I would recommend to add them in the graphs.

O. In Fig 4D there are two asterisks above the standard deviation bars although an asterisk is already shown underlining the significant differences above an ad-hoc horizontal bar.

P. The differences in the immunocytochemistry of Fig 5D are not easy to see, perhaps the authors can slightly enlarge the figures and adjust the pictures to improve visibility.

Q. In page 14, second paragraph, after the sentence "Three of these sites have been identified as Ago2-bound in the human CLIP experiments" it would be adequate to add the corresponding citation.

R. In the first sentence of second paragraph in page 17 (Discussion), the sentence ends with a reference "73" which is evidently a mistake, since citations in EMM do not have that format. The same occurs in the first sentence of the last paragraph of the same page that refers to a "Piwecka paper".

S. Fig 1F-G, what the arrows (up in F down in G) means? Nothing is described in the corresponding legend.

Answer to Reviewers:

Referee #1 (Remarks for Author):

The manuscript 'A CircRNAs Resource indicates an AGO2-related regulatory role for circSLC8A1 in the Parkinsonian brain' represents a thorough attempt to identify circRNAs that are functionally related to the pathophysiology of Parkinson's disease. Notably, the authors make use of a high-quality collection of postmortem human brain samples, namely the one of the Netherlands Brain Bank in order to draw disease-relevant conclusions for the human PD brain. CircRNAs are emerging as a ncRNA species with a wide array of regulatory roles in the healthy and diseased CNS. Nevertheless, the understanding of their cross-interactions with the degenerating brain is currently limited, and studies like the one reported here can provide a valuable resource of information that could be then used for independent validation.

Thank you for your positive evaluation of our research topic and work.

However, the manuscript in its current form, lacks several pivotal elements of appropriate analysis and interpretation, as discussed below.

Major remarks:

- One of my greatest concerns is the fact that the authors try to interpret data derived from bulk tissue RNA-seq, without acknowledging the limitations of such an approach and the confounding variables that it may introduce into the analysis of the data. Ideally, cell type deconvolution tools should be used to address any issues of this sort.

We thank the reviewer for bringing this up. We included cell deconvolution in the original manuscript (Figure 1F and 1G), but the details were not clear enough. Following this comment, we added a clearer explanation in the methods section and in the results.

- Related to the previous point, how can the authors know whether the lower circRNA levels in PD SN are not caused by neuronal loss? Instead, the authors attribute this observation to a particular PD-specific regulatory mechanism. This should be addressed.

As a matter of fact, we do attribute much of the observed changes to the disease-induced neuronal cell loss, especially at the substantia nigra where such loss has been recognized for a while now. Some of those changes could be direct (i.e. loss of expression of a given RNA) and some indirect. Indeed, our findings indicate that many different cell types, not only neurons have been affected by the disease, as one would expect following a massive loss of neurons in a specific brain region; hence the mentioning of PD-specific regulatory mechanism. In any case, following the comment raised by the reviewer, we revised the text to highlight the fact that such changes may be secondary to the neuronal cell loss.

- Alu editing index does not significantly differ between CTR and PD SN. This suggests that RNA editing is not mechanistically relevant to PD pathology. The authors should discuss. How could then the differential expression of circSLC8A1 in PD SN (compared to CTR) be explained?

We thank the review for bringing this up as it was not clear in the original MS. We agree that Alu editing does not significantly differ between CTR and PD SN. In response to this reviewer's comments, we repeated our editing analysis, which essentially confirmed that observation; and we revised the section of the text discussing that point to explain this outcome more clearly. This being said, we do not attribute the emergence of circSLC8A1 to modified editing, but rather to altered splicing events; as it is now clearly stated in the revised text.

- No causal relationship between circSLC8A1 and miR-128 can be supported by the current data. The authors should either tone down the pertinent passage or address this issue by assessing the levels of miR-128 in vitro following transfection of circSLC8A1.

Comment accepted. To address this point, we performed knockdown cell culture experiments of circSLC8A1 followed by RNA-seq analysis. Interestingly, we found that knock down of circSLC8A1 resulted in modified expression of certain miR-128 targets (see Supplementary figure 7 in the new version of the manuscript). In any case we have tuned down the statements all along the manuscript.

- The statistical tools used for each experiment should be clearly indicated. What type of statistics was used in Figure 4D?

Thank you for this comment. The statistical tools are now specified for each experiment in the figure legend, including Figure 4D.

Other remarks:

- The authors should provide the gene sets used for pathway analysis.
Provided as requested, added as a separate Supp data file.

- The levels of all microRNAs indicated in Figure 6E should be assessed.
As stated in the revised manuscript, the levels of these miRNAs were looked at in various web-available datasets, the details of which are noted.

- It is not entirely clear how only samples with RIN values above 6.5 were used for library preparation (as indicated in the text), while in the pertinent Sup. Table1, 42 samples do not fulfill those criteria.

We thank the reviewer for bringing this up as it was clearly confusing in the MS. We only performed RNA-seq on the samples with RIN values above 6.5, as our analysis indicated that these RIN values ensure independence of the RNA-seq outcome on the RNA quality (see Barbash et al., Neurobiol. of Disease 2017). However, the remaining samples have been used for qPCR validation tests, as is now clearly noted in the revised Methods.

- The direction of pathway enrichment (repressed/induced) should be indicated for all GO term enrichment analyses.

Done and presented (Supp data files for detailed genes used for GO terms and detailed GO analysis results). Thanks!

- The terms 'upregulation' and 'downregulation' should be used with caution taken that any

apparent gene level changes may be driven by differences in cell numbers in the bulk analysis.

Thanks, noted as requested.

- The authors should provide average values (and corresponding statistics) for the within region variability of their CTR vs PD cohorts (analyzing differences of gender, age, Braak stage, Amyloid stage, PMI).

Done and presented in Supplementary table 1.

- PMI values should be provided for all samples.

Done and presented in Supplementary data files.

- Fold change, p-value and corrected p-value for all identified circRNAs should be included.

Done and presented (please see new Supplementary Data file).

- There are several typos.

Sorry about that; typos corrected.

Referee #2 (Remarks for Author):

Hanan and colleagues performed RNA-seq to identify changes in circular RNA expression in the brains from controls and donors with Parkinson's disease. Overall trends are described, including suggestions about circRNA biogenesis via changes in Alu editing levels. The authors then focus on circSLC8A1 which is upregulated in the PD SN. A model is proposed in which circSLC8A1 binds miR-128 and functions as a sponge. The manuscript is dense but generally clear and I suggest a few additional analyses.

Thank you for this positive assessment of our work.

Major point:

(1) My main concern is about stoichiometry. What is the relative ratio of circSLC8A1 to miR-128 in cells? They should be at similar levels in order for the circRNA to function as a sponge. A related point that the authors should address: what is the relative amount of linear vs circular RNA generated from the SLC8A1 gene?

This is indeed an important point. As we stated in the MS, the effects of circSLC8A1 on miRNA function might be more complicated than simple degradation. We have tuned-down some of the statements and discussed this at more depth. We agree this is an important point but believe the exact mechanism of regulation goes beyond the scope of the present manuscript. We have included a few sentences stating this in the new version of the manuscript. We thank the reviewer for bringing this up.

(2) Fig 6G/H: The authors show 3 miR-128 targets that increase in PD brains, but these are cherry picked mRNA targets. What percentage of miR-128 targets increase in PD brains? If the authors' model is correct, most direct miR-128 targets should show increased expressed levels.

We indeed get the point raised by the reviewer. As stated above, we don't know how miR-128 activity is altered but our results suggest that perturbation of circSLC8C leads to miss-regulation of miR-128 targets. While it is true that we validated the miss regulation of a few miR128 targets, we also found that miRNA-128 targets were enriched far more than expected by chance among the DE expressed genes in the PD brains (Fisher exact test statistic $p=0.0036$, Figure 6I). In addition, we have now performed and analyzed an RNA-seq profiling experiment of cells in which circSLC8A1 is knocked down. We indeed observed that 24 out of the 110 up regulated genes were predicted miR-128 targets, also more than expected by chance (Fisher exact test, $p=0.0433$) as is now shown in the revised manuscript. So, while our evidence is not 100% conclusive and does not address the exact mechanism of action, it suggests a path for further exploration. We have now stated this more clearly in the discussion.

Minor points:

(1) Last sentence of abstract is over-sold. The current manuscript does not provide

experiments that suggest a preventive value for modulation of circSLC8A1. If the authors want to make such a claim, they need to show changes in cell survival etc when circSLC8A1 expression is experimentally modulated.

Comment accepted; we have toned down that statement.

(2) P.2: "However, it remained unclear what are the consequences of this binding (Piwecka et al)" is an odd phrase because the CDR1as mouse model described in the cited paper does suggest a mechanism.

True, text modified to reflect that.

(3) P.3: The authors may also wish to mention that the presence of multiple Alu elements in introns can allow alternative circularization events.

Done as requested.

(4) Supplementary Fig 2G should have error bars to show the variation in expression levels across samples.

Done and presented. Thanks for noticing.

(5) Page 7: When describing the circRNA profiling, it would be helpful to more clearly clarify the criteria used. What is a circRNA expressed at "very low levels"? How many sequencing reads must be observed for a circRNA to be annotated? Do the circRNAs need to be detected by both algorithms? What was the degree of overlap in circRNA predictions between algorithms? Some of these points are addressed in the methods but they would be helpful to also have in the main text.

We thank the reviewer for bringing this up. We have now included more details in the text, methods and figure legends.

(6) The callouts for Supplementary Fig 3B and 3C are reversed on page 7.

Sorry about that, we corrected it.

(7) Fig 2A: I find the model to be confusing as drawn. Back-splicing does not lead to a hairpin shaped transcript. The hairpin shaped transcript occurs first and then back-splicing occurs.

Thanks for this comment, we have modified the drawn model accordingly.

(8) Fig 3C: The authors should comment on why they think editing levels decrease in the SN and why reduced levels of circRNAs are observed. One would have expected increased circRNA levels as is seen in the MTG and AMG.

Thanks for this comment; we believe that the reduced levels of circRNAs as well as editing changes in the SN might reflect loss of neurons in this brain region, but don't have a conclusive explanation. In any case we have added this possibility in the revised text.

(9) Page 9: "Surprisingly, we did not observe positive correlation in any of the other two assayed tissues" - Please show these data.

Done and presented.

(10) Figure 4A could be more informative, e.g. what exon(s) are included in the circRNA? Is it flanked by Alu repeats? A related question: is editing observed at the SLC8A1 locus?

We thank the reviewer for bringing this up as we agree it was confusing in the initial version of the manuscript. To address the first point, we have added a little more detail to the figure legend. Indeed, circSLC8A1 is formed by only the 2nd exon of the gene. Regarding the second point, there are no intronic inverted repeats in the proximity of the circularizable exon (at least 5000 bases up or downstream). We have added this information to the new version of the manuscript. Last but not least, we did not find any ALU element in the circRNA (and neither in the 1000 and 5000 bp windows surrounding it, and even the wider 10000 window contains only a single element). In an effort to solve this difficulty, we used the RNA editing tool on the full regions (not just the Alu elements). Unfortunately, the results were so noisy that we could not tell whether the signal is real or not. Therefore, we did not add Alu-editing results specifically for circSLC8A1 in this part of the manuscript.

(11) Figure 4D: Please provide statistics for mRNA between CT and PD. What is the measured p-value?

We performed the statistical test and found no significant differences. We have added the statistics and p-value to the Supp table 3.

(12) Figure 4E: Please provide statistics.

Done and presented.

(13) Page 13: When referring to the published Ago2-CLIP data, please make clear that one cannot distinguish binding to linear vs circRNA unless the CLIP reads span the back-splicing junction.

Thanks for this comment. We agree on the importance of making this clear. Indeed, we detected the back-splicing junction itself as AGO-2 may bind one of the recognition sites. We have modified the text to make this clear (see Supp table 4 for the sequence found in Ago2-CLIP).

Referee #3 (Remarks for Author):

The report of Hanan et al. entitled CircRNAs Resource indicates an AGO2-related regulatory role for circSLC8A1 in the Parkinsonian brain describes for the first time a catalog of CircRNAs (together with mRNAs and microRNAs) present in three brain regions derived from Parkinson disease patients and healthy controls and points towards circSLC8A1 as an Ago2-associated circRNA, regulated by oxidative stress, and abundant in Parkinson Disease (PD)-derived substantia nigra that influences mir-128 activity.

Thanks for this succinct summary of our work.

Major findings/novelty of the manuscript:

This is the first large and systematic study linking circRNAs to PD. For doing so the authors applied a tour de force approach and sequence 3 brain regions in dozens of control and PD individuals. They find that circRNAs globally change in the brain of PD individuals (in all the 3 regions studied but with opposite results between the SN and the other 2 regions). This is a key and relevant finding and surprisingly it is not highlighted enough in the manuscript (not even in the abstract).

Thanks for this valuable comment, we were apparently too shy about our work and have now highlighted its novelty and importance in the revised abstract.

Moreover, the authors establish a link between circRNAs and PD by finding global changes of circular transcripts in different brain regions. They describe a differentially expressed set of circRNAs in the substantia nigra (SN) of PD individuals. Additional relevant findings are i- the pronounced brain region-specificity of gene expression patterns, ii- and the age-dependent circRNA accumulation in the human brain. Finally, authors focus on one highly expressed CircRNA in PD material and perform an initial molecular characterization.

This is certainly a timely and relevant resource paper. It is based on a dataset generated directly from a large and well-curated set of human brain material derived from PD patients and healthy controls. Additionally, the massive sequencing on which the Kadener's lab has an extended experience and the different ways to present and analyze the data add additional value to the work. Having said that, there are numerous points which, though they do not put under question the main findings of the manuscript might certainly contribute, at least in the eyes of this reviewer, to a more clear presentation and interpretation of the data.

Comment appreciated; we have now made a focused effort to present our data and interpret its implications more clearly.

Main

Comments

1. The writing is in many passages too convoluted and difficult to follow. The text will greatly benefit from conceptual and language simplification. In addition, the authors tend to overstate their findings, which is not necessary and sometimes deviates the attention of the reader. For example: There is no clear indication that circSLC8A1 has a causative role on PD so, it is not necessary in the abstract to indicate that the results "advocate a preventive value for its modulation".

We thank the reviewer for the valuable advice. We have now edited and simplified the language in the manuscript.

2. Another issue of relevance is related to the main and running titles of the manuscript. The main title suggests a causal ("regulatory") role of AGO2-related regulation of circSLC8A1 in PD and the running title states advocates for a "modulatory" effects of circSLC8A1 via Ago2. This is mainly a resource paper and these titles sound close to overstatements since a causal role of the circular form of SLC8A1 is actually not proven in the manuscript. Taking this into account I would suggest a title that more strictly described the main findings of the study.

Thanks; we have revised the title as recommended.

3. The authors should better describe why they selected MTG and AMG as additional brain areas. Actually, both brain structures are involved in the control and implementation of stress-related programs and anxiety-like behaviors what makes them also particularly interesting per se, but also in the context of PD since anxiety is also an often associated symptom in PD patients. Along this line, and considering that the present study is intended to be a resource article, the authors should invest more efforts to describe in more detail the catalogue of mRNA and circRNAs identified not only in the SN but also in MTG and AMG brain regions both in PD and control samples. For instance, which are the circRNAs present in the amygdala of healthy and PD individuals? I don't find this kind of data clearly described in the paper, neither in figures nor in tables and I have the feeling the paper would greatly benefit from precisely described and list the candidates from these brain areas.

We agree, and performed text revision as recommended while respecting the journal's space limitations, tables with Supplementary information added for MTG and Amygdala.

4.a) Regarding the presentation of the sequencing data: almost all data in the paper seems to be focused on up-regulated candidates but downregulation is equally relevant and should also be clearly described both in text and figures. The term differentially expressed (DE) genes is not accurate enough. For instance, the data presented in supplementary table 3 might be organized dividing up-regulated and down-regulated and not merely by FDR. Moreover, the co-regulation data presented in Fig 1H-J is constructed from up-regulated genes? what about the existence of down-regulated modules?

Comment accepted and data presented more clearly, with an emphasis on up-and down-regulated transcripts. The correlation analysis (WGCNA) includes sets of genes that share the same pattern of expression, so it includes both upregulated and down-regulated genes.

b) And regarding the co-regulated modules, I don't find which are the genes that constitute the modules. If I am right, the authors should describe them in a supplementary figure; at least the most significantly different modules.

Comment accepted, and the requested supplementary data added to the revised manuscript (list of all genes in each module, WGCNA module genes).

c) In the particular case of circRNAs, it would be interesting if the authors could add a major level of detail to the presented data, for instance, the distribution of sizes, whether the circRNAs are intronic vs exonic, which types of regions they contain (5' UTR, ORF, 3' UTR), number of exons or exact junctions, etc.

Thank you for this comment. The requested analysis of circRNA is added to the paper as Supplementary file circ coordinates analysis. We understand that is a good idea to

present a deeper analysis of the found circRNAs. However, we feel that several papers have deeply characterized human brain circRNAs, so we found it a little redundant and feel that will make the paper more unfocused.

5. In page 7 the authors described that "as much as 26% of the detected circRNAs but only 2% of the mRNAs were unique to the SN (Figure 2E, grey), whereas 19% of the identified circRNAs but 82% of the mRNAs were shared between all tissues (Figure 2E, green)." This is very important finding. It should be underlined and the authors should elaborate more about it in the discussion.

Done as recommended.

6. In page 7 second paragraph the authors state that "...the SN expressed higher total numbers of circRNAs compared to the MTG and the AMG (normalized to library total reads, Figure 3A," This is very interesting finding that raise the question as to whether the SN also express higher levels of mRNAs transcripts too. Consequently, how about the comparison of the expression levels of their linear counterparts? Do they inversely correlate?

The comparison shows no correlation between circRNA and mRNA counterparts, similar to other papers describing circRNA expression changes and no correlation to the host gene mRNAs. When comparing mRNA expression of SN and other issues we could not detect overall elevated in gene expression. It is important to remember the limitations of the method we used since we are comparing on average the same number of reads from each sample so elevation in RNA content in the SN would be very difficult to measure.

7. In Fig2C, the data described includes both healthy controls and PD? Again, the authors should rather show the data about differentially expressed RNA species in each separate brain regions comparing healthy and PD patients.

Figure 2B shows the differences between control and PD samples, we wished to show the more prominent tissue specificity in 2C, enhanced by the heatmap in 2D.

8. The data presented suggest that the circSLC8A1 somehow regulates miR128 but it is not clear how (and it is likely beyond the scope of this study), so the authors should limit to state with caution how the circRNA might modulate the miRNA.

Thank you for this comment. To address this point, we have now characterized the gene expression changes upon knock down of circSLC8A1 in cells in culture. Indeed, we observed a significant change in the mRNA levels of a subset of miR-128 targets, as is now shown in the revised manuscript. In addition, we saw that miR-128 targets are enriched among statistically-significantly upregulated genes (FDR<0.05) in the PD SN. In any case we followed the advice of the reviewer and have toned down the miR-128/circSLC8 link along the manuscript.

9. As circSLC8A1 seems to be upregulated in PD and by oxidative stress, it would be interesting to know whether oxidative stress globally induces transcription from the locus or induces circularization of the exons. This could be easily done in cell culture by qPCR from intronic sequences upon exposure to PQ. Moreover, the authors should look for inverted repeats or potential RBP binding sites. Any finding (even if no sequences are found) would be informative.

As suggested by the reviewer we looked for inverted repeats 5Kb up and downstream the circularizable exon and could not find any. We have added this information to the

new version of the manuscript. Regarding the proposed experiment, it is indeed a good idea, but we believe it is beyond the scope of the present manuscript.

Minor points

A. If the submission format allows it, in case a putative new version of the manuscript should be reevaluated, I would thank the authors the addition of the following features to the submitted document:

Page numbers

Line numbers

Accommodate figures and their corresponding legends together

We did our very best and appreciate the added value to our manuscript thanks to your meticulous comments and suggested changes. The figure legend separation was a request from the instructions of the journal.

B. The legends are often so concise that basic information to properly understand the figures is missing.

Legends were revised and expanded in the revised manuscript.

C. In page 7 last sentence authors write: "This suggested a particular regulation and maybe importance of circRNAs in the PD SN"

This reviewer wonders whether the differences could be at least partially due to neuronal loss. The authors should mention and discuss this possibility.

That is what we cautiously refer to in the discussion of the revised manuscript.

D. In page 9 one can find the following sentence "We conclude that RNA editing and RNA circularization are anti-correlated in PD and control brains." I am confused with this statement because this anticorrelation seems to be lost in the SN because the editing does not change in that area. Please clarify.

We clarified the corresponding figures and text accordingly in the revised manuscript. Briefly, we show for individual samples that there is anti-correlation between circRNA expression and editing levels (Fig 3D). Indeed, we saw this trend when we looked at circRNA expression averages both in each tissue and in CT/PD comparison, and when we averaged the editing levels in the same manner. An exception was the SN PD, where we could not see the expected elevation. We suspect that this is due to neuronal loss since neurons are the main contributors to editing events in the brain. We hope that now this is clear and thank the reviewer for bringing this up.

E. In page 9 second paragraph the authors claim the following regarding the age-dependent accumulation of circRNAs "Surprisingly, we did not observe positive correlation in any of the other two assayed tissues, but this could be due to the limited age range of the assayed samples".

I don't find it described in figures. Is this a "not shown" data? In my view, it should be shown.

We agree with the reviewer; it is now shown in the revised manuscript in Supplementary information.

F. In page 9 the third paragraph starts as follow: "We then look for differentially expressed (DE) circRNAs in the brain tissues from PD and healthy individuals. We indeed identified 24 DE circRNAs (corrected p value < 0.05) between control and PD tissues (Figure 3H)."

Is this a pooled data emanating from all brain regions together or just from SN? Please clarify and in case the first case, please explain why.

Thanks; this refers to the pooled data from all regions, as is now stated in the revised manuscript.

G. In Fig 3H what are the red dots? Significant DE candidates? This is a relevant figure. It might be important to clarify this and provide a clear-cut table with those significant (and especially relevant) circRNAs. In addition, the short names of those genes might also be written in the figure, perhaps making the figure a bit larger.

Yes, the red dots are statistically significant DE circRNAs, which is added to the figure legend now. Comment appreciated and revision performed: table of these circRNAs added to Supp information.

H. In page 13 at the end of first paragraph the following sentence "However, the levels of this circRNA remained unchanged in control and PD fibroblast samples (Schulze, Sommer et al., 2018) (Supplementary Figure 6C). This might indicate that under normal growth conditions, the regulation over circularization and therefore the balance between circSLC8A1/SLC8A1 expression does not change, even in the case of genetic PD background."

Comment appreciated and unclear sentence revised as recommended.

I. Immediately after, one can reads "The latter result suggests that changes in circSLC8A1 expression in the PD brains might be related to other aspects of PD like oxidative stress." I am not clear which "other aspects" the authors really mean since the previous sentence is referring to differentiation process of fibroblast as starting material, which are not strict "aspects of PD". The sentence is unclear to me. Perhaps the authors mean that the changes in circSLC8A1 might be secondary to cellular insults or challenges reported to occur in dopaminergic neurons such as oxidative stress. If so, this should be formulated in a more clear way.

Correct- this is precisely what we thought, and sentence revised as suggested.

J. In the Fig 4I only the blots are shown. A quantification of the blots would be desirable.

Quantification added as suggested

K. In data described in supplementary figure 6C is somehow difficult to follow since the legend provides almost no details. Which kind of sample each column represents? Is this graph showing previously published data mixed with original data form the paper? Unclear to me, but if so, that should be clearly described and stated. On the other hand, the title of Supplementary Fig 6 mentions the use of ES cells, not iPS Cells, which kind of cells have been actually employed in this figure?

Thank you for the comment. Supplementary Fig 6B includes original data and presents experiments where we used ES cells (H9) and differentiated them to explore the expression of circSLC8A1 and SLC8A1 mRNA during embryonic differentiation. Sup. Fig 6C is a published dataset of mRNA expression by Schulze, Sommer et al., 2018 (hence we couldn't measure circSLC8A1 expression).

L. Many citations are wrongly described. Multiple times only the start (but not the end) page is written (e.g. Gal-Mark et al, Grunner et al, Holdt et al., Langfelder et al, Min et al., etc), In many other cases no pages are described at all (e.g. Agarwall et al or Piwecka et al.,). On the other hand there are no spaces between citations what makes difficult the rapid finding of the citations. Please correct these mistakes.

Our apologies; references corrected as suggested, except for those issues which followed the journal's instructions (e.g. spacing between citations).

M. In page 7 the mention of figure 3B and 3c seem to be interchanged.

True, many thanks for noticing. Corrected.

N. Some graphs with Cartesian axes have major ticks but most of others graphs do not have. The ticks help the reader to more precisely evaluate the graphs; I would recommend to add them in the graphs.

Comment accepted and revision performed as suggested.

O. In Fig 4D there are two asterisks above the standard deviation bars although an asterisk is already shown underlining the significant differences above an ad-hoc horizontal bar.

Thanks for noticing, these two asterisks are actually outliers in the corresponding boxplots.

P. The differences in the immunocytochemistry of Fig 5D are not easy to see, perhaps the authors can slightly enlarge the figures and adjust the pictures to improve visibility.

Done as recommended

Q. In page 14, second paragraph, after the sentence "Three of these sites have been identified as Ago2-bound in the human CLIP experiments" it would be adequate to add the corresponding citation.

Indeed, added as suggested

R. In the first sentence of second paragraph in page 17 (Discussion), the sentence ends with a reference "73" which is evidently a mistake, since citations in EMM do not have that format. The same occurs in the first sentence of the last paragraph of the same page that refers to a "Piwecka paper".

Our apologies, revised

S. Fig 1F-G, what the arrows (up in F down in G) means? Nothing is described in the corresponding legend.

Up/down arrows indicate gene groups that were up/down regulated and separated in the analysis. Legend revised to improve clarity, thanks again for devoting so much attention to our work.

28th May 2020

Dear Prof. Kadener,

Thank you for the submission of your revised manuscript to EMBO Molecular Medicine. We have now received the enclosed reports from the referees that were asked to re-assess it. As you will see the reviewers are now globally supportive and I am pleased to inform you that we will be able to accept your manuscript pending the following final amendments:

1) Please address the minor text changes commented by referees 1 and 2 and add the requested data, including performing the stoichiometry experiment (ref.2).

Please provide a point-by-point letter INCLUDING my comments as well as the reviewer's reports and your detailed responses to their comments (as Word file).

2) Address all editorial requirements.

Please submit your revised manuscript within two weeks.

I look forward to reading a new revised version of your manuscript as soon as possible.

Yours sincerely,

Celine Carret

Celine Carret, PhD
Senior Editor
EMBO Molecular Medicine

*** Instructions to submit your revised manuscript ***

To submit your manuscript, please follow this link:

Link Not Accessible

***** Reviewer's comments *****

Referee #1 (Remarks for Author):

The authors addressed the majority of my remarks on the previous version. However, some additional adjustments are necessary before the manuscript could be considered for publication at EMM. I understand this is a complex dataset and the authors have put extra effort to functionally validate their findings. Nevertheless, the authors' claims are still at times exaggerated and do not reflect accurate interpretation of their data. Correlative evidence should be explicitly stated as such when applicable.

More specifically:

1. The following is a highly speculative statement. It should either be eliminated or drastically modified.

'Therefore, dopaminergic neurons may be under elevated risk of neurodegeneration, and the balance between circularization and canonical splicing may change their survival ability under diverse insults, especially if excessive oxidation tilts this balance towards circRNA production. While possible, this mechanism might be restricted to a small subset of cells, as we don't see significant changes on SLC8A1 mRNA.'

2. Similar adjustments should be made in the last sentence of the following passage:

'However, the SN of PD patients showed lower circRNA numbers than in the SN of healthy controls (T-test $p=0.02$, Figure 3A), which could be attributed to and be secondary to the neuronal cell loss in the SN of PD patients and/or to altered splicing events in this brain region. This suggested a particular regulation and possibly importance of circRNAs in the PD SN'

3. Regarding the in vitro knockdown experiment:

- The authors should provide all the genesets of differentially expressed transcripts
- The same for DE genes that are miR-128 predicted targets (both in the upregulated and downregulated subsets)
- Which miRNA prediction algorithm was used for defining miR-128 targets?
- How many miR-128 predicted targets are included in the downregulated genes?

Pending on these additional remarks, the authors should adapt their conclusions.

Finally, the authors should include page number references to their text when addressing their responses to reviewers' comments.

Referee #2 (Remarks for Author):

The authors have largely addressed my concerns. I suggest one additional analysis to measure circRNA vs microRNA stoichiometry. The other comments are all clarifications to the writing.

(1) Supp Fig 7: The authors should determine the stoichiometry of miR-128 vs circSLC8A1. The other data in the figure look very promising, but the stoichiometry measurement is critical for the author's overall model about sponging. In lines 584-586, they write this is an issue for the future, but it will be very easy to do in the 293 cells.

Other comments:

(1) Manuscript title: I feel the authors have tried to include too much information in too few words.

(2) Line 291: Cite figure rather than stating "Supplementary material"

(3) Figure 4D: In the main text, the authors refer to the left and right panels for circRNA and mRNA respectively but this is not how the data are displayed.

(4) Figure 4G: This panel suggests that circRNA levels are decreasing and that this is responsible for the lack of correlation between mRNA and circRNA levels. The prior panels showed that the circRNA level is increased so the authors should explain how both results are true.

(5) Line 355: These results could also be from oxidation preventing circRNA degradation.

(6) Supp Figure 6E: Please label Ago2 band.

(7) Line 394: Clarify to say there are 11 Ago2 binding sites, not CLIP binding sites.

Referee #3 (Comments on Novelty/Model System for Author):

All these issues have already been commented in the original submission review and they remained unchanged for this reviewed version.

Referee #3 (Remarks for Author):

I am happy with all the modifications the authors introduced in the manuscript, which in my view has improved significantly. They have addressed almost all the issues I raised and I think the manuscript is ready to be published in EMBO Mol. Med.

The authors performed the requested editorial changes.

***** Reviewer's comments *****

Referee #1 (Remarks for Author):

The authors addressed the majority of my remarks on the previous version. However, some additional adjustments are necessary before the manuscript could be considered for publication at EMM. I understand this is a complex dataset and the authors have put extra effort to functionally validate their findings. Nevertheless, the authors' claims are still at times exaggerated and do not reflect accurate interpretation of their data. Correlative evidence should be explicitly stated as such when applicable.

More specifically:

1. The following is a highly speculative statement. It should either be eliminated or drastically modified.

'Therefore, dopaminergic neurons may be under elevated risk of neurodegeneration, and the balance between circularization and canonical splicing may change their survival ability under diverse insults, especially if excessive oxidation tilts this balance towards circRNA production. While possible, this mechanism might be restricted to a small subset of cells, as we don't see significant changes on SLC8A1 mRNA.'

We have modified this statement in page 12 line 537. Thanks.

2. Similar adjustments should be made in the last sentence of the following passage:

'However, the SN of PD patients showed lower circRNA numbers than in the SN of healthy controls (T-test $p = 0.02$, Figure 3A), which could be attributed to and be secondary to the neuronal cell loss in the SN of PD patients and/or to altered splicing events in this brain region. This suggested a particular regulation and possibly importance of circRNAs in the PD SN'

We have eliminated this last sentence in page 6 line 249. Thanks.

3. Regarding the in vitro knockdown experiment:

- The authors should provide all the gene sets of differentially expressed transcripts

Added as Dataset EV 9, page 10 line 447.

- The same for DE genes that are miR-128 predicted targets (both in the upregulated and downregulated subsets)

Added as Dataset EV 10, page 10 line 453.

- Which miRNA prediction algorithm was used for defining miR-128 targets?

Diana prediction tool was used for target prediction, (<http://diana.imis.athena-innovation.gr/DianaTools/index.php>), added to materials and methods, page 17 line 755.

- How many miR-128 predicted targets are included in the downregulated genes?
Pending on these additional remarks, the authors should adapt their conclusions.

Thank you for your comments, out of DE genes in the knock-down experiment of circSLC8A1, we detected 10 miR-128 targets that were reduced (out of 99 down-regulated genes). Calculating the same statistical test as for the upregulated genes (The Fisher exact test, as calculated in page 10, line 428 in the manuscript) gave us the p-value is 0.7523. Therefore, we are satisfied with the current conclusions in the text.

Finally, the authors should include page number references to their text when addressing their responses to reviewers' comments.

Done, thank you for this suggestion.

Referee #2 (Remarks for Author):

The authors have largely addressed my concerns. I suggest one additional analysis to measure circRNA vs microRNA stoichiometry. The other comments are all clarifications to the writing.

(1) Supp Fig 7: The authors should determine the stoichiometry of miR-128 vs circSLC8A1. The other data in the figure look very promising, but the stoichiometry measurement is critical for the author's overall model about sponging. In lines 584-586, they write this is an issue for the future, but it will be very easy to do in the 293 cells.

Thank you for your suggested experiment. We performed this stoichiometry measurement and calculated that for the qPCR cycles we detected for circSLC8A1 and miR-128 in SN and in SH-SY cells we found the following measurements: 6.6aM for circSLC8A1 and 34.21aM for miR-128, with 7 binding sites for miR-128 in circSLC8A1, resulting in 0.73 as the miRNA/circRNA sites ratio). We believe that this ratio in expression could result in sponging effect for miR-128 by circSLC8A1, with full calculations in Dataset EV8, page 9 line 415.

Other comments:

(1) Manuscript title: I feel the authors have tried to include too much information in too few words.

We have changed the title of the manuscript

(2) Line 291: Cite figure rather than stating "Supplementary material"

Thanks, changed to Appendix sup. Figure 2.

(3) Figure 4D: In the main text, the authors refer to the left and right panels for circRNA and mRNA respectively but this is not how the data are displayed.

Thanks, these references were corrected to the color of the bar (purple or grey), page 7 lines 317-318.

(4) Figure 4G: This panel suggests that circRNA levels are decreasing and that this is responsible for the lack of correlation between mRNA and circRNA levels. The prior panels showed that the circRNA level is increased so the authors should explain how both results are true.

Thanks for bringing this up, indeed the axis are different, and this contributes to the confusion. Indeed, these are the data used for the Figure above (qPCR) and does indeed show increased levels of the circRNA. We have now added a sentence to the figure legends for avoiding this confusion. See change in page 27, line 1181.

(5) Line 355: These results could also be from oxidation preventing circRNA degradation.

This comment was added to the manuscript as an alternative explanation, page 8 line 353.

(6) Supp Figure 6E: Please label Ago2 band.

Thanks! Added to the figure and figure legend. This figure is now Appendix S3 figure

(7) Line 394: Clarify to say there are 11 Ago2 binding sites, not CLIP binding sites.

Corrected, Thank you for the clarification, see correction in page 9 line 384.

Referee #3 (Comments on Novelty/Model System for Author):

All these issues have already been commented in the original submission review and they remained unchanged for this reviewed version.

Referee #3 (Remarks for Author):

I am happy with all the modifications the authors introduced in the manuscript, which in my view has improved significantly. They have addressed almost all the issues I raised and I think the manuscript is ready to be published in EMBO Mol. Med.

Thank you!

25th Jun 2020

Dear Prof. Kadener,

We are pleased to inform you that your manuscript is accepted for publication and is now being sent to our publisher to be included in the next available issue of EMBO Molecular Medicine.

Please read below for additional IMPORTANT information regarding your article, its publication and the production process.

Congratulations on your interesting work,

Celine

Celine Carret, PhD
Senior Editor
EMBO Molecular Medicine

Follow us on Twitter @EmboMolMed
Sign up for eTOCs at embopress.org/alertsfeeds

Corresponding Author Name: skadener@brandeis.edu and hermona.soreq@mail.huji.ac.il;
Journal Submitted to: EMBO molecular medicine
Manuscript Number: EMM-2019-11942